

# A survey of Chinese interpreting studies: who influences who … and why?

Ziyun Xu[1] and Leonid B. Pekelis[2]

[1] Universitat Rovira i Virgili, Spain
[2] Stanford University, USA

## ABSTRACT

This paper describes how scholars in Chinese Interpreting Studies (CIS) interact with each other and form discrete circles of influence. It also discusses what it means to be an influential scholar in the community, and the relationship between an author's choice of research topic and his academic influence. The study examines an all-but-exhaustive collection of 59,303 citations from 1,289 MA theses, 32 doctoral dissertations and 2,909 research papers, combining traditional citation analysis with the newer Social Network Analysis to paint a panorama of CIS. It concludes that the community cannot be broadly divided into Liberal Arts and Empirical Science camps; rather, it comprises several distinct communities with various defining features. The analysis also reveals that the top Western influencers have an array of academic backgrounds and research interests across many different disciplines, whereas their Chinese counterparts are predominantly focused on Interpreting Studies. Last but not least, there is found to be a positive correlation between choosing non-mainstream research topics and having a high level of academic influence in the community.

Corresponding author
Ziyun Xu, xuziyun@gmail.com

## INTRODUCTION

The earliest documentary record of interpreting dates back as far as 3000 BCE—the Ancient Egyptians had a hieroglyph for it (*Delisle & Woodsworth, 1995*)—but it can be assumed reasonably safely that the first interpreters started work as soon as cavemen realized they could not be sure to make themselves understood by neighboring tribes using gestures and signs alone. Given its extremely long history, it is somewhat surprising that it only became an independent field of academic enquiry in the 1990s, when scholars began consciously to use the term Interpreting Studies, to distinguish it from the original 'parent' Translation Studies. Despite the mystique which still surrounds the profession to a certain degree,[1] since the Second World War interpreters have been increasingly in demand to bridge communicative divides wherever they might arise—war crimes tribunals, peace-keeping operations, high-level international trade negotiations, low-level sightseeing trips … the list is endless.

[1] Until the release in 2005 of the movie *The Interpreter*, starring Nicole Kidman, many an outsider was no doubt unsure of the difference between written translation and spoken interpreting.

Chinese interpreters came to prominence on the international stage when the People's Republic of China (PRC) regained its seat at the United Nations (UN) in 1971. As a result of China's return the UN was instrumental in establishing a dedicated training program to meet the demand for conference interpreting services from various of its offices all over the world (*Wang, 2006*). The first research article on Chinese Interpreting Studies (CIS) archived by CNKI[2] was published in the late '50s (*Tang & Zhou, 1958*), and since then the discipline's growth has been explosive: a total of over 3,600 scholars have to date produced nearly 3,000 journal articles and conference proceedings, 1,300 MA theses and over 30 dissertations on the subject. Given its rapid evolution and ever-heightening academic status, it is of crucial importance to study the structure of this scientific community. The purpose of the present scientometric survey is to marry the traditional technique of citation analysis with the newer one of Social Network Analysis (SNA) to obtain a fuller picture of the ways in which CIS scholars communicate with each other both formally and informally.

[2] The China National Knowledge Infrastructure is by far the nation's most comprehensive academic database, archiving conference proceedings, journal articles, MA theses and doctoral dissertations dating back to the early 1900s.

## MAJOR QUESTIONS

To gain an understanding of how scholars in CIS communicate with one another to generate learning and advance the field, both citation analysis, which describes formal networks of influence (*Baumgartner & Pieters, 2003*), and social network analysis, which identifies informal communities (*Otte & Rousseau, 2002*), have been used in this study. Using an all-but-exhaustive collection of citation data, we ask how authors interact, how we can characterize who is influential, and what being influential means. These questions have long attracted attention in the scientific community (see for example *Van Dalen & Henkens, 2001*; *Haslam et al., 2008*; *Buela-Casal et al., 2009*; *Chen & Redner, 2010*; *Ravallion & Wagstaff, 2011*), partly in consequence of *Kuhn*'s (*1970*) seminal work on the nature of science, in which he emphasizes the importance of adopting a data-driven approach to analyzing the structure of the scientific community, and partly because academic authorities are ever vigilant to ensure that the investments they make in their researchers' projects are justified by the results—that they are, in short, spending their money wisely.

## LITERATURE REVIEW

The growth of scientometrics in China has trailed developments in the West by about a decade. In an article in *People's Daily* in *1977*, Hsue-Shen Tsien, a scientist influential in the development of missiles and space programs in both China and the United States, argued for the need to establish an independent discipline that focused on the 'science of science.' Hongzhou Zhao is considered the pioneer of scientometric research in China. While working at a labor camp in Henan Province in 1974, he analyzed the *History of Natural Science* published by the Fudan University Journal, studying the way in which research production centers 'shift' from place to place around the world over time (*Liu, 1999b*). In 1985, he and Gouhua Jiang published in *Scientometrics* an article on the demographics of scientists—this was the first time a Chinese-authored article on the subject received widespread recognition from the international community (*Zhao & Jiang, 1985*). In 1978,

the Research Group for the Science of Science became the first academic body in China to study scientometrics (*Jiang, 2008*).

China developed its own citation indexes, though they appeared much later than in the West and are still not fully comprehensive or standardized. The leading ones include the state-funded Chinese Science Citation Database (CSCD), created in 1989, and the Chinese Social Sciences Citation Index (CSSCI), established in 1998. Unlike its Western counterpart, the Social Sciences Citation Index (SSCI), the CSSCI collects data from monographs, collective volumes and miscellaneous websites in addition to journal articles. However, while the SSCI includes articles dating back to 1972, the Chinese equivalent contains those published only since 1998, seriously limiting the pool that researchers can draw on for analysis. As an example, a search for 'Interpreting' in the CSSCI currently yields only 263 entries; by contrast, the same search of SSCI produces 585.

Despite the difficulty in accessing data, multiple scientometric studies across various academic disciplines have been carried out in China (see for example, *Zhang & Zhang, 1997*; *Wang et al., 2005*; *Ruan, 2012*). A handful of researchers have applied the principles and methods of scientometrics to CIS. For example, some have attempted to provide a broad overview of trends and developments in the discipline by classifying relevant journal articles by theme and giving a few examples of the leading articles in each category (*Hu & Sheng, 2000*; *Liu & Wang, 2007*; *Li, 2007*). Others have gone further by backing up their claims with simple counts of articles published on a given theme (*Mu & Wang, 2009*; *Tang, 2010*).

*Gao (2008)* and *Zhang (2011)* have published studies of the similarities and differences between translation and interpreting research in China and in the West. Whereas Zhang takes a broad view, covering the entire subject of interpreting, Gao's interest is in the cognition aspect of Simultaneous Interpreting Studies.[3] The latter's corpus of articles includes ones from eight leading Western academic journals[4] published between 2000 and 2007, and ones published in three premier Chinese journals[5] between 1994 and 2007. Her analysis suggested that recent works in the West showed fewer signs of debate over the nature of interpreting than were common previously, and that instead they frequently drew on findings from contemporary psychology to re-evaluate prior research in the field. Gao noted the strong influence on CIS of late-20th-century major Western theories such as Gile's Effort Model, cognitive pragmatics and the Interpretive Theory, and observed that, unlike their colleagues in the West, Chinese researchers were less inclined to interdisciplinarity.

## THE PRESENT STUDY

There is clearly a growing interest in scientometrics among CIS researchers. Aside from Gao, most authors to date have applied its more basic methods and principles, mainly article counts, in their research. This is the logical place to start, but there is ample room to employ the more complex approaches scientometrics offers. Doing so will shed light on the true impact made by individual scholars, and will provide other more finely tuned information about the evolution of specific fields of research in a relatively objective way.

[3] This sub-discipline deals with the various aspects of how interpreters manage to render a speaker of one language's meaning into another at the same time as they are speaking.

[4] The Western journals are: *Interpreters' Newsletter, Interpreting, Meta, The Translator, Babel, Hermes, Target* and *Forum.*

[5] The Chinese journals are: *Chinese Translators Journal, Chinese Science Technology Translators' Journal* and *Shanghai Journal of Translators for Science and Technology.*

As (*Lowry, Karuga & Richardson, 2007*) point out, mere numbers do not permit a nuanced analysis of influences within a given discipline; for example, an author may have published numerous articles but have little influence among his peers. In addition, a small sample population may cause significant biases and affect the outcome of any analysis conducted. The present research study is intended to contribute to CIS by carrying out, for perhaps the first time, a thorough scientometric survey of the literature, including journal articles, theses and dissertations. Its aim is to provide scholars with a comprehensive and objective overview of the interactions between scholars in the field, and of which academics are the most influential and how their choice of particular subjects of enquiry relates to their impact on research as a whole.

## Research questions

Expanding on the major questions outlined at the beginning of this paper, three specific research questions were developed to ascertain how CIS scholars interact with each other and how influence is defined in the community. The following section outlines the authors' rationale for investigating each research question.

**1. How do CIS authors interact with one another? Do they form defined communities? If so, what are the features of those communities?**

The advancement of a science relies heavily on its participants' communicating and collaborating with one another: scholars build on each other's research, and work together to address common issues or to replicate colleagues' experiments under different conditions to investigate whether their conclusions can be extrapolated to a larger population. In the context of citation analysis, identifying community structures can help us understand the predominant research themes in a given field and how certain subject matters grow or decline in popularity over time.

**2. In terms of citations, who are the most influential scholars in the CIS literature?**

*Nederhof (2006)* observed significant differences in citation behavior between natural and social sciences: members of the former communities (physics, chemistry, etc.) tended to influence each other across geographical boundaries, whereas those of the latter (sociology, linguistics, etc.) generally had very limited influence beyond the countries in which they lived. One might think Interpreting Studies would be an exception to this rule: because of its focus on the interactions between languages and cultures, its authors might reasonably be expected to exert influence across the boundaries of language, geography and culture. Identifying the most influential scholars in CIS ought to reveal whether any Western researchers have an impact on the Chinese field and if so why; it should also prove useful for identifying differences between the most influential Western and Chinese authors' backgrounds, and for exploring the dominant schools of thought in CIS.

**3. Are there any research topics that influential scholars tend to write about? What themes and keywords correlate with author influence?**

All researchers would like to see their papers frequently cited by others, and hope that their colleagues might be inspired to pursue their work and address any questions that may remain unanswered. However, the reality is that few articles published are highly

influential: a far higher number are rarely read or cited by others. Studying what makes an article influential is a useful exercise from three perspectives: firstly, every scholar would like to make a mark within his research community, and so would do well to know what makes for a successful paper; secondly, because the bodies which allocate grants and other forms of funding always want to be assured that their investments are money well spent (generally speaking, scholars need to publish when they receive a grant); and lastly, studying these predictors of influence can help to identify the hottest topics in the field. A handful of researchers have already explored the issue of what makes one article influential and another not. *Buela-Casal et al. (2009)* examined the relative influence of theoretical and empirical papers in three Spanish psychology journals, concluding that the former type received twice as many citations as the latter. *Haslam et al. (2008)* studied the citation data for 308 articles in social-personality psychology, and found the following factors particularly strong for predicting an article's impact: (1) the reputation of the first-listed author; (2) the presence of a senior colleague's name among a new author's collaborators; and (3) a journal's ranking. The present study is the first time that the predictors of influence in CIS have been analyzed.

## Data collection and organization

Given the paucity of coverage of CIS citation data in existing academic databases, for the present study a near-comprehensive database of 59,303 citations was built from scratch—they represent citations from the 1,289 Chinese MA theses, 32 doctoral dissertations and 2,909 research papers available to the authors. These three bodies of literature, chosen because they best represent the overall state of CIS, were accessed through multiple channels: field trips to university libraries, interlibrary loans, book purchases, and academic databases such as CNKI, Wanfang and the National Digital Library of Theses and Dissertations in Taiwan—thus ensuring wide coverage of academic works in both mainland China and Taiwan. Once collected, the references were manually entered into Excel Spreadsheets, using the idea behind Structured Query Language (SQL) to managing data. The method employs multiple interactive and cross-referenced tables; in the present case there are three such: Documents, Authors, and Citations. For these tables to interact with one another it is important to have unique 'keys,' elements that allow each row to be identified: a unique, consistent 'author key,' for example, enables us to know whether a particular person in the Authors table is the same as one found in a row of the Documents table.

The key concept behind the present analysis is that of the citation network: the documents are nodes in this network, with arrows interconnecting them when authors cite their predecessors (see Fig. 1). The nodes are labeled with various attributes such as 'author,' 'publication year,' and 'keywords.' The number of nodes associated with an author is the number of documents he has produced. Authors share a node when they have co-authored a document. Arrows are in the direction of the citation, so the cited work is at the receiving end of the arrow. When arrows are used to indicate the presence of a citation between documents, there can be at most one arrow between each pair of

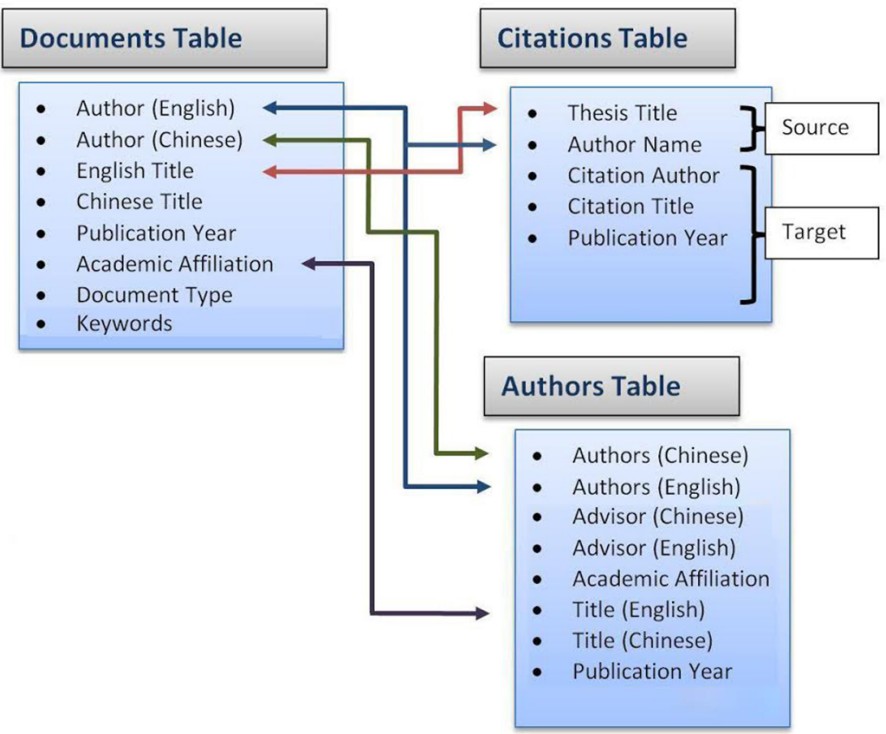

**Figure 1 Data organization framework using the idea behind SQL relational data management system.** Arrows between tables show examples of unique pieces of information which are indexed in multiple tables, as a way of linking data points between tables.

nodes; when they are used to indicate a type of citation, there can be multiple arrows, as one document may cite another multiple times. A total of ten Excel Spreadsheets were compiled: three documents table for CIS doctoral dissertations, MA theses and academic papers; six citation tables—CIS doctoral dissertations (English and Chinese citations), MA theses (English and Chinese citations), and academic papers (English and Chinese citations), and one author table. This data format makes it easy to be exported into Tulip for data visualization (see section 'Author interaction') and 'R' for correlational analysis (see section 'Research topics and academic influence'). The raw data alongside the source codes has been deposited into GitHub, and can be accessed from this link: https://github.com/danielxu85/CIS.

## Description of topic labeling method

Rather than rely solely on the keywords provided by the papers' authors, the content of each and every publication in the data-set was carefully analyzed with the aim of generating keywords that best captured the topics they covered. The keywords typically chosen by authors can often be too general or specific and mask hidden trends. The keywords tagged for the present study were further grouped into six broad themes: Training, Professional, Language, Socio-cultural, Cognitive and Miscellaneous. This classification system was adapted from the coding scheme used in *Gile*'s (*2000*) study—it covers all the issues addressed in CIS and minimizes overlap between categories. It should be noted, though,

that each paper under examination may be tagged with multiple themes depending on its content. Consolidating keywords into themed categories can help identify major trends in CIS which might otherwise have been drowned out by the sheer number of keywords generated.

# METHODOLOGY

## Author interaction

Using near-exhaustive citation data, the present authors wished to determine whether the communities of CIS are best classified according to the hypotheses put forward by earlier scholars (*Moser-Mercer, 1994*; *Gile, 2005*; *Gile, 2013b*). In the first two of these studies, it was noted that there existed two dominant and opposing camps—those who approached research from a liberal arts standpoint and those who leaned towards natural/empirical science—and that there was little communication between the two. In his later study, Gile refined his original classification by suggesting that Translation & Interpreting Studies can be divided into three broad groupings—Human Sciences Culture, Canonical Scientific Culture, and Practice, Reflection, Generalization.

To address this aspect of CIS, visual representations of its citation networks were created. A number of open source software applications are available for analyzing and visualizing networks: some, such as SciMAT and VantagePoint, are well suited for analysis, while others (e.g. Pajek and Gephi) are better for visualization. Tulip, a program designed for analyzing and visualizing relational data, provides a good balance because it incorporates a number of layout and clustering algorithms, in addition to network metrics. More importantly, Tulip was selected over other applications for its capacity to visualize large complex networks—the complete citation network used in the present study contains over 12,000 nodes and in excess of 50,000 edges.

Tulip enabled us to import the entire CIS citation network data in the form of a list of all edges (connections between nodes) in the data-set. Once this was done it gave the option to apply a number of layout algorithms. In the present instance, once the entire CIS citation network had been imported, we selected a force-directed algorithm—Fast Multipole Multilevel Method (FM$^3$)—to lay out the graphs presented in the images seen in Figs. 3 and 4 (*Hachul & Jünger, 2005*). This algorithm groups nodes that are multiply connected to each other closer together in the layout while distancing ones that are not directly connected. This approach makes the produced images ideally suited for visually detecting community structures in networks.

A panoramic graph of the research interactions between various CIS authors was created using the 2012 citation data. In this graph, vertices represent authors and the edges represent the number of citations between them. To generate a full network image with Tulip software, a placement algorithm was used to lay out the nodes. The edges were rendered invisible to ensure that the nodes could be seen clearly. The node color was set to blue using the property option available in Tulip.

To study the relationships between the most influential nodes in the network, we firstly filtered out the less important ones, retaining only the top 150, then calculated

the PageRank score of each that remained. The PageRank algorithm for citation networks measures the importance of an author by gauging the quality and quantity of other authors that cite him. The author whose PageRank is being calculated ('the target') is said to be at a distance of 0 from himself, and each author who cites him at a distance of 1. The importance of these Distance 1 authors is in turn calculated by the quality and quantity of authors citing them, who are at a distance of 2 from the target. The underlying idea is that influential authors will be much cited by other influential authors, while non-influential authors will either not be cited or only be cited by other non-influential authors. The process continues to a pre-determined parameter known as the damping factor. In line with the majority of citation studies, which use an average chain length of 10 to 15, for the present study a damping factor of 0.15 was used.

PageRanks were calculated using the Algorithm-Measure-Graph-PageRank option in Tulip. The top-ranking 150 nodes were selected, and an induced sub-graph[6] was created which showed only these 150. Again, the FM$^3$ algorithm was used to lay out the nodes, and Edge Bundling was used to merge edges which were close to one another to make the layout more readable. Tulip automatically colors nodes as soon as any metric has been calculated. In the present instance, the nodes were colored using a gradient scale from blue (high) to orange (low) values. Since it was not possible to visualize the names of all the authors, we chose to display only nodes with a high value of any metric, which in this case was PageRank.

In addition to visually identifying the communities within CIS, a quantitative analysis was conducted to verify whether the network had two or more clusters. Community detection pinpoints the most natural groupings of individuals present in a network (*Schaeffer, 2007*). There are a number of ways to evaluate the quality of such groupings, one of the most widely used in recent times being Q Modularity, which was introduced by Newman in 2004.

Modularity is defined as the percentage of all connections that fall within a community, minus the expected percentage of connections in the community. The expected percentage is based on the assumption that connections are distributed completely at random, with no regard for community structure. The modularity cut algorithm determines how to partition a graph into communities with high modularity scores, hence the communities it identifies all contain a higher number of intra-community connections than might be expected to occur purely at random. Modularity values lie between 0 and 1; higher values are desirable and represent better clusterings. Typical values lie in the range 0.3 to 0.7 (*Newman & Girvan, 2004*), values between 0.0 and 0.2 suggesting that the graph is entirely random with no known community structures (*Rényi & Erdős, 1959*).

Seven commonly used network clustering algorithms were used to determine the number of clusters and optimize the process of clustering CIS citation data; Spin-glass stood out as the best at defining CIS communities. Modularity values, which were obtained for different numbers of clusters using the Spin-glass algorithm, indicate the optimum number of clusters for CIS.

[6] An induced sub-graph is one that highlights only a certain number of nodes and all the edges connecting them, but omits all the other nodes and non-connecting edges.

Finally, to analyze the common features of the discrete communities of CIS, the image of clusters was generated. The Spin-glass algorithm was applied with seven clusters. Subgraphs of each cluster were generated with a unique color for each different cluster. FM$^3$ was used to layout the nodes of each cluster individually by executing the layout algorithm on each of the subgraphs. Again, the edges were rendered invisible to make the graphs more readable, and nodes were labeled with author names. PageRank was employed to identify important nodes and labels were displayed only for the most important from each cluster.

It should be emphasized here that modularity cut does not involve the use of meta-information about authors to determine how they are divided into communities. Previous researchers in SNA have not attempted to use meta-data about each citation to generalize the features of each community, because a variety of factors can drive authors into a certain community: a connection can be established between two authors because they have co-authored an article together, or because they have cited one another's research, a process known as co-author citation (*Newman, 2001*). In addition, meta-information, such as the content of each cited paper and background information regarding each cited author, cannot be obtained from commercial databases, which means manual labor is required to obtain and screen thousands of papers. *Newman & Girvan (2004)* applied the modularity cut algorithm to identify communities based on co-authorship data: authors were added to the network as vertices, edges between them indicating their co-authorship of one or more papers in the data-set. Newman and Girvan were primarily interested in investigating whether researchers from the same community were acquainted with one another. *Takeda & Kajikawa (2010)* analyzed citation data in the fields of energy and material science by tracking modularity scores obtained from each clustering iteration, but they stopped short at summarizing the features that define each community: clustering only indicates that the nodes in each group are similar, but the similarity is dependent upon whether the nodes are connected, not why they are connected.

## Most influential scholars

All the Western and Chinese authors appearing in the citation data were ranked according to their degree centrality (DC)[7] and weighted degree centrality (WDC)[8] measures, and to their PageRank Algorithm (PRA) scores. DC and WDC are first-order centrality measures, while PRA is a higher-order measure. The three highlight different aspects of an author's influence within a network: for example, out-degree and weighted out-degree measures indicate how well authors disseminate information about the work of other scholars, while PRA assigns them scores based on how well connected they are with influential colleagues.

It was expected that some authors, despite publishing prolifically, would be found to have little research impact within the CIS community, while others, despite publishing very little, may nonetheless be widely cited. There were also grounds to expect that the ways in which influential authors are cited may be subject to variation: they might be highly influential thanks to a wide range of much-cited work, or their influence might depend on a small number of seminal works. In addition, the present author ventured to predict that,

[7] Degree centrality calculates the number of edges connected to a particular node in the network. It has two sub-categories: an author's in-degree centrality represents the number of other authors citing him, while his out-degree is the number of others cited by him.

[8] Weighted degree centrality is the total number of citations an author makes and receives. Weighted in-degree centrality is the total number of citations of his work made by others, while weighted out-degree centrality is the total number of citations he makes in all his publications.

despite the homogenous background of CIS researchers (*Zhang, 2008*), authors from other disciplines may have topped the influence polls.

## Research topics and academic influence

There are various methods for measuring an author's influence, such as DC, WDC and PRA (see Question 2 above), but for this analysis only in-degree centrality, out-degree centrality, PRA and EigenVector centrality were used. WDC was excluded because it and DC are both first-order centrality measures: they essentially measure the same aspect of an author's influence. The present study's database contains 2,909 research articles, 1,289 MA theses and 32 doctoral dissertations; from these were extracted 978 unique keywords to describe their contents. All the keywords for each author in the database were tallied up, and a keyword profile created for each, representing the relative frequency with which he or she used a keyword, normalized to 1, i.e., the number of times that keyword was used divided by the total number of keywords he or she used—hence a keyword with a value of 0.3 represents 30% of all keywords used by that author. Normalization was added in to prevent there being undue emphasis on the connection between keywords and centrality measures when identifying prolific authors.

Such a large number of keywords has two limitations. Firstly, many of them are synonyms or hypernyms, leading to conceptual overlap. Second, in any regression the larger the number of explanatory variables, the more data they require in order to maintain statistical power—the ability to detect significant relationships between explanatory variables and the response (see for example *Hsieh, Bloch & Larsen, 1998*). In the present case the explanatory variables, namely the relative frequencies of the almost 1,000 keywords, were too numerous in relation to the number of documents for the 'bulk' statistics to yield good results. To avoid this sizable stumbling-block, the keywords were classified into six themed groups: Cognitive, Language, Professional, Socio-cultural, Training and Miscellaneous. The categories were designed to be non-overlapping so as to allow for the drawing-out of meaningful trends which would otherwise be undetectable amid the crowd of keywords present in the documents. A supplementary analysis of the 978 keywords, giving a more fine-grained picture of the data, is reported at the end of this section.

The authors' network measures (in-degree centrality, out-degree centrality, PRA and EigenVector centrality) were first matched to their theme profiles, which came from separate databases: journal articles, MA theses, and PhD dissertations. Of the 2,277 journal article authors, 1,023 were matched in the network database; of the 1,289 MA theses authors, 1,092 were matched; and finally, of the 32 PhD authors, 29 were matched. In total, roughly 60% of authors in the theme profiles database were matched with network measures. Following the same methodology, a subsequent mapping of keywords to themes was also performed.

The reason that 40% of the authors could not be matched in the network influence database was that no references were available in their works. The majority of these authors did not include any bibliography in their papers, and the works of a very small

proportion—mainly authors of theses—are embargoed by their affiliated institutions. This is an inherent property of the data, and it is an important component to document when describing the customs of different academic cultures. Because of China's unique intellectual traditions in the early stages of CIS' development as an academic discipline, the overwhelming majority of papers published had no bibliographic references. In addition, many early papers were case studies of the authors' own experiences, and their documentary nature precluded the need for many citations. However, these early studies were included in the data-set for two reasons. Firstly, these articles were produced during CIS's initial stage, as per *Schneider*'s (*2009*) model of the development of scientific disciplines, and to exclude them would be to miss out on a significant portion of the early literature. Secondly, many of them received citations from later studies, which indicated that they served as the foundation for the development of CIS and brought academic value to the field.

To investigate how well themes act as predictors of the network influences of CIS authors, the most simple approach was linear regression; this is a good starting point for becoming acquainted with the data and is typically used as a first step in statistical analysis for examining the null hypothesis that the explanatory variables have no relationship at all to the response variable. For the preliminary analysis, separate linear regression models were fit with theme profiles as the predictors, using DC, WDC, PRA and EigenVector as a response variable in each model. Since each theme profile was normalized, authors who published numerous papers and those who published only a few had similar-looking theme profiles, therefore the number of papers published by an author was added as a predictor. An F-test was used to determine if any of the regression models could explain variation in author influence in statistically significant terms. The F-test results showed that the linear regressions did not explain the variations in author influence very well. High levels of disparity between authors' influences was the suspected reason for this. To confirm this suspicion, the disparity was calculated by means of the Gini Coefficient, which measures the concentration of mass in a cumulative distribution, and is borrowed from the field of economics.

A linear model is a simple approach but it makes strong assumptions about the relationship between the response variable and the explanatory variables, therefore it came as no surprise that the linear regression failed to explain the data well. The next approach adopted was one which, though less ambitious in terms of explanatory power, is far less dependent on assumptions. This alternative approach involved dividing network measures into three groups ('bins'): high-, middle-, and low-ranking. However, rather than assuming that these three bins were equally probable and so spacing their cutoffs regularly, a data-based approach was employed to determine where they should fall. A total of 20 cutoff points were considered, corresponding to the percentiles of network measures, from 0 to 95 in steps of 5. Creating the three groups called for finding specific lower and upper cutoffs for the middle-ranking group. An example of where the cutoffs between the three bins might be placed would be at the 5th percentile between the low- and middle-ranking groups, and at the 85th between middle- and high-ranking (see Fig. 2).

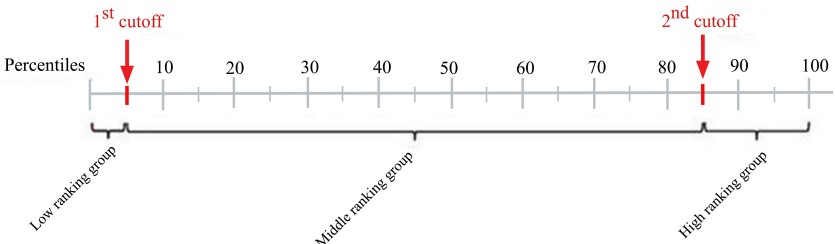

**Figure 2 Ranking group cutoffs.** 5% of all authors are in the low-ranking group, 15% in the high-ranking, and the remaining 80% in the middle-ranking.

The effect of an author's theme profile on the probability of his belonging to each of the three groups was estimated by means of a multinomial regression. For each of the different groups, a statistical analysis known as a deviance test was performed to assess whether its divisions were good for explaining authors' influence based on their theme profiles and number of publications. This procedure allows a numerical quantity known as the $p$-value to be calculated. For each model, the smaller the $p$-value, the more likely the model is a good fit.

Given that there were 190 different combinations of cutoff points to consider for each response variable, we could not simply report as significant those cutoff points that had a $p$-value of less than 0.05. If we did, by the definition of a $p$-value, we could expect about 10 cutoff points to be significant even if there was no relationship between authors' meme profiles and influence for any cutoff point. Therefore, a statistical procedure was used to find cutoff points that gave considerable evidence of a good model fit over and above the fact that we were choosing from 190 different models. This procedure can be measured by a quantity called False Discovery Rate (FDR), which is defined as the expected proportion of false discoveries, or cutoff points that are not significantly related to authors' theme profiles, from all cutoff points detected as significant. In statistics, the FDR can be controlled by using the Benjamini–Hochberg procedure. The smaller a group of models' FDR, the greater their chance of representing true underlying effects rather than random variation in the data. The procedure assigns to each model a score called the $q$-value. One way of interpreting this value is that to build a group of models with a certain given maximum FDR, only those models with a $q$-value below that FDR should be included. Consequently a model is considered good enough for inclusion in a group if it has a small $q$-value.

Several hypotheses were formulated before the statistical analysis was conducted. One such was that scholars typically perform literature searches by submitting keywords to search engines that rank results from the most recent to the oldest. Under this hypothesis, it was expected that authors writing on commonly studied subjects would, because of sheer weight of numbers, have difficulty becoming highly influential. Conversely, authors writing about rarely-studied subject matters would be far more likely to receive attention from colleagues tackling the same subjects, translating into numerous incoming citations for them. Another reasonable hypothesis was that authors might use other methods of performing literature searches, such as finding citations in existing papers or sorting results

based on relevance rather than how recently the items were published. The analytical methodology described in this paragraphs is an important first step towards testing the veracity or otherwise of these hypothetical scenarios.

To examine whether CIS authors' full keyword profiles were significantly correlated with any of the network measures, one additional analysis was performed—regularized regression. The assumption made at this point was that the majority of keywords were not highly correlated with influence, while a small minority were. In statistics this is called a sparsity assumption (*Hurley & Rickard, 2009*). Had a simple linear regression of measure of influence been run on keyword profiles, we would have expected to obtain a large number of very small regression coefficients (one for each unique keyword), some medium-sized, and maybe a few large ones. Adding too many non-significant terms into a standard regression would have obscured the signal from significant terms, hence the need to use regularized regression for removing non-significant keywords. Regularized regression addressed this issue by zeroing out many of the insignificant coefficients. More specifically, a regularization technique called Lasso was run for multinomial regression (*Tibshirani, 1996*) with 10-fold cross-validation to approximate the optimal set of non-significant keywords and set their coefficients to 0. The remaining keywords were considered to be significantly correlated with the network measure. Similar outcomes were hypothesized from the keyword profile analysis as from the earlier theme analysis. Some of the frequently used keywords were expected to be correlated with the low influence group, whereas some rarely used ones were expected to be correlated with high influence. The reason for this predicted outcome was the same as the one for themes described in the previous paragraph: authors whose papers have unique keywords are more likely to be read and cited by fellow researchers than those with common keywords.

## RESULTS AND DISCUSSIONS

### Author interaction

The nodes representing authors were situated in Fig. 3 using FM$^3$. Contrary to the expectation that the field of CIS is composed of polarized camps which barely communicate with one another, Fig. 3 suggests rather that its scholars cannot be easily divided into clearly separable communities. In addition, the degree distribution of the entire CIS graph follows a scale-free behavior, which implies that several nodes with high In-Degree and Out-Degree scores perform the function of holding the graph together: these nodes are to be found at the center of Fig. 3, those with lower In-Degree and Out-Degree scores being pushed towards the periphery. To corroborate this finding, the citation patterns of the top 150 CIS authors were also visualized (see Fig. 4). The ranking of these authors was determined using the PageRank algorithm on the entire network. If there were to be well-defined communities, they would have been clearly visible, but in reality Fig. 4 also illustrated a hairball effect, which means that even the top-ranking authors closely cite each other's works.

This finding suggests that CIS researchers do not form opposing camps marked by a distinct intellectual preference for liberal arts or empirical sciences, and that certain

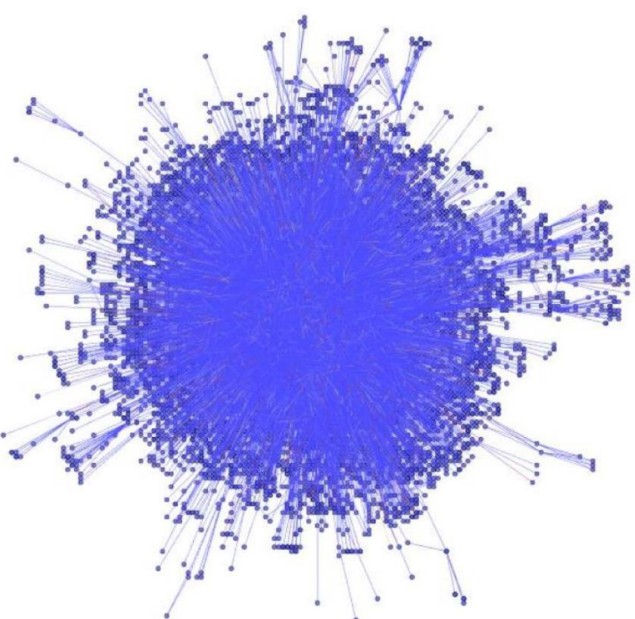

**Figure 3 CIS citation network visualization.** Entire CIS network laid out using a force-directed algorithm.

influential scholars are well-cited across the board. This is in sharp contrast to the situation in WIS, where for personal and professional reasons authors may choose not to include certain items of relevant research in their works: scholars from ESIT, France's most famous IS institute, almost exclusively cite one another's work and avoid research by scholars from disciplines outside interpreting; the latter reciprocate by ignoring the interpreting specialists' work, adjudging it to be 'unscientific' (*Gile, 1999*).

To investigate CIS community structures in greater depth, a number of leading network-clustering algorithms were examined; the purpose of these was to optimize the clustering process and determine the optimum number of clusters. Table 1 shows the modularity values that resulted from testing each algorithm using CIS citation data to cluster the network. Spin-glass yielded the highest modularity value with seven clusters, so was adopted for this study (*Fortunato, 2010*). Another reason for using this algorithm is that it allows the user to input the number of clusters required by specifying the number of spins in the system. Other clustering algorithms do not generally allow the number of clusters to be used as input as they determine the number of clusters by optimizing some objective function or by optimizing some dynamic process.

The Spin-glass clustering algorithm is based on spin models, which are popular in statistical mechanics (*Reichardt & Bornholdt, 2004*). The underlying idea is based on the principle that nodes connected to each other should belong to the same cluster and ones not directly connected should belong to different clusters. If Potts spin variables are assigned to the vertices of a network, and if the interactions are between neighboring spins, structural clusters can be found from spin alignment of the system. Spins of nodes within clusters are similar and different across clusters with the purpose of maximizing Potts energy.

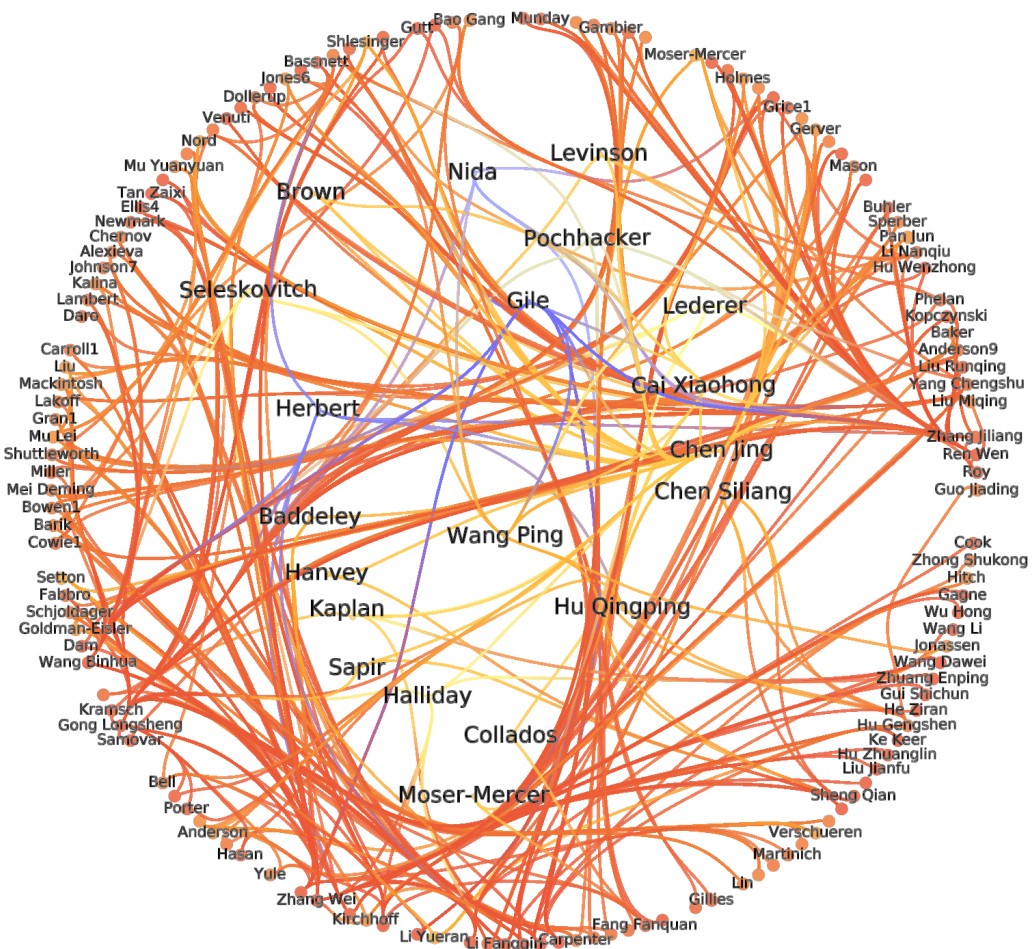

**Figure 4 Top 150 CIS authors.** Studying the behavior of interconnections of the top 150 authors. Nodes are placed using a circular algorithm where the most important nodes are displayed in the center to highlight top ranked authors.

**Table 1 Comparison of different clustering algorithms using CIS data.**

| Clustering algorithms | Number of clusters generated | Modularity |
|---|---|---|
| Louvain | 20 | 0.416 |
| FastGreedy | 87 | 0.387 |
| Walktrap | 693 | 0.266 |
| Leading eigen vector | 2 | 0.241 |
| Infomap | 643 | 0.339 |
| Label Prop | 76 | 0.062 |
| Spin-glass | 7 | 0.434 |

Figure 5 displays the seven clusters obtained when the Spin-glass clustering algorithm was applied to the CIS citation data network. Nodes in each of these clusters were assigned their positions using the aforementioned FM$^3$ algorithm. Their closely-packed appearance represents the dense structure of these communities as they exchange frequent citations

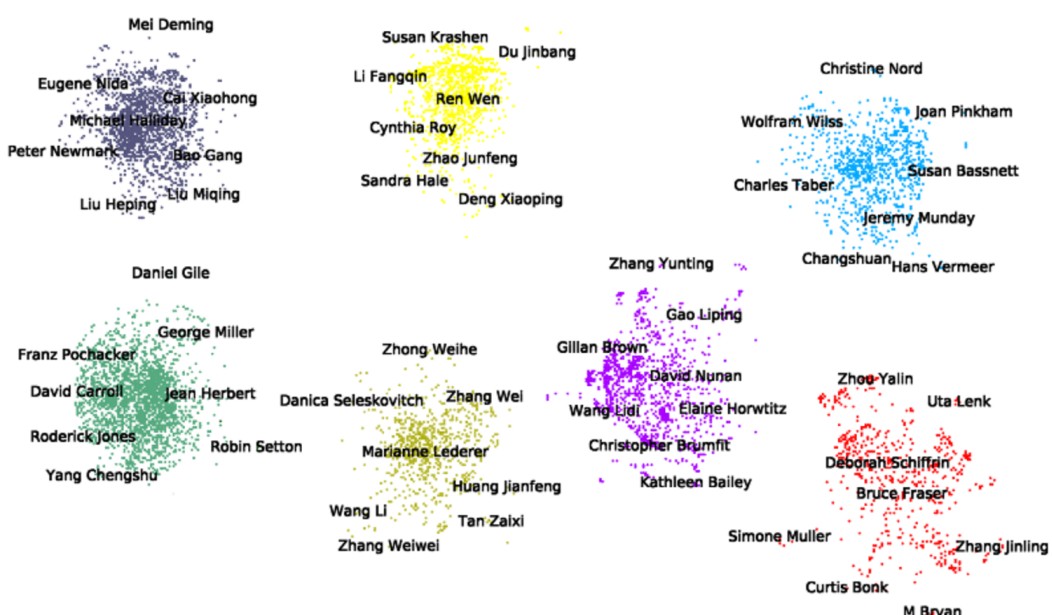

**Figure 5 CIS communities.** Seven communities in CIS obtained by the Spin-glass algorithm.

with other members of the same community. The names of the influential authors in each cluster are displayed to make it easy to identify the principal communities the network can be subdivided into. This was achieved by applying the PageRank algorithm to the entire network and ranking the importance of each node based on its PageRank value.

Daniel Gile had the highest PRA scores in the green cluster, the largest community in the data-set with 2,366 scholars. Authors such as Franz Pöchhacker, Jean Herbert, Roderick Jones, Robin Setton, Yang Chengshu, David Carroll, George Miller, John Andreson also belonged to this cluster. Most of these top influencers' works are concerned with cognitive psychology or psycholinguistics, which would appear to indicate that those disciplines have a dominant influence on CIS research. Most notably, George Miller's *The Magic Number Seven, Plus or Minus Two* (*Miller, 1956*), and John Anderson's *Cognitive Psychology and Its Implications* (*Anderson, 1980*), received 103 and 106 incoming citations, respectively, suggesting that the interdisciplinary approach to interpreting research is one favored by many a scholar. It should be noted here that this community features outliers, but further research would be required to ascertain why, alongside scholars whose particular focus is cognition, CIS authors cite writers such as Herbert and Jones, neither of whom specializes in cognitive science, though both have produced popular textbooks on interpreting techniques.

Only one of the top ten scholars in this cluster was Chinese—this was Yang Chengshu, who has served as director of the Graduate School of Translation and Interpretation at Fu Jen Catholic University in Taiwan since 2006. She received a total of 195 incoming citations from CIS scholars, her two most often-cited works being *Research into Interpreter Training* (*Yang, 2005*) (In-Degree = 240) and *Types of and Rules for Syntactic Linearity in SI* (*Wan & Yang, 2005*) (In-Degree = 2). Her high-ranking in this cluster—6th out of 2,366

scholars—is worthy of remark: given that Taiwan has a somewhat different socio-cultural background from that of mainland China, and that her specialty is Japanese–Chinese interpreting, her great popularity among CIS scholars was unexpected. Several factors may explain this: though Yang has worked exclusively at academic institutions in Taiwan, she received her PhD in Interpreting Studies from BFSU under the supervision of Wang Kefei in 2006. She also has close ties with scholars from the PRC, evidenced by the fact that the second of her most frequently cited works mentioned above was co-authored with Wan Hongyu, a scholar from Shanghai. In addition, the popularity of her other most-cited work indicates that the general principles of interpreter training know no linguistic boundaries.

## Most influential scholars

The following 30 people were identified as the most influential Western scholars in CIS based on their PRA scores (see Table 2).

Gile, Lederer, and Seleskovitch head the table, indicating that their research is highly influential among their Chinese colleagues. When it comes to degree centrality (DC) measures the ranking has shifted slightly: Gile, cited by 1,077 Chinese scholars, continues on top, while Seleskovitch, cited by 756, now ranks second. She is trailed by Lederer, who was cited by 674 researchers. Their ranking by weighted degree centrality (WDC) remained consistent with their DC ranking: Gile scored 2,116, Seleskovitch 1,207 and Lederer 1,020. The first had a total of 75 papers and books receive incoming citations. The most widely cited of his works was *Basic Concepts and Models for Interpreter and Translator Training*, whose main contribution to the field is theoretical, though it does have a number of empirical components (See Table 3). Seleskovitch and Lederer formulated the Interpretive Theory of Translation, which stresses the importance of disregarding the linguistic structure of the original when interpreting into a foreign language. Despite both having published empirical research, almost all Chinese scholars exclusively cite the translations of their non-empirical works such as *Interpreting for International Conferences: Problems of Language and Communication* (*Seleskovitch, 1978a*), *Interpréter Pour Traduire*, and *Pédagogie Raisonné de l'Interprétation*—these were among the earliest Western interpreting theories introduced into China. Their influence on CIS research is perhaps attributable to the wide availability of the translated versions of their works. Upon further analysis, of the top 30 Western researchers, their works alone were cited in translation; all the others were cited in the original language—for example, though two of Gile's French papers received incoming citations, they were both cited in French. The citation data revealed that 27 of Seleskovitch's works and 23 of Lederer's received citations. Tables 4 and 5 list the top five most cited works of these two authors.

Being cited by a large number of scholars does not necessarily translate into a particular author's having a high PageRank score or being perceived as highly influential by his confreres. For instance, David Gerver was cited by 160 Chinese scholars and received 257 citations, ranking 20th in terms of DC and 16th in WDC, but his ranking dropped to 40th when calculated by the PRA. *Gile*'s (*2000*) paper indicates that Gerver's work is also reasonably popular in the West. There is no doubt that Gerver was an influential pioneer in

**Table 2  The top 30 Western scholars in CIS by PRA measures.**

| No. | PageRank | Author's name | In-degree | Out-degree | Degree | Weighted degree | Weighted in-degree | Weighted out-degree |
|---|---|---|---|---|---|---|---|---|
| 1 | 0.00773432 | Daniel Gile | 1077 | 0 | 1077 | 2116 | 2116 | 0 |
| 2 | 0.00496506 | Marianne Lederer | 674 | 0 | 674 | 1020 | 1020 | 0 |
| 3 | 0.00487847 | Danica Seleskovitch | 756 | 1 | 757 | 1208 | 1207 | 1 |
| 4 | 0.00251794 | Eugene Nida | 467 | 0 | 467 | 730 | 730 | 0 |
| 5 | 0.00230936 | Franz Pochhacker | 538 | 0 | 538 | 903 | 903 | 0 |
| 6 | 0.00169282 | Roderick Jones | 277 | 0 | 277 | 310 | 310 | 0 |
| 7 | 0.00158975 | Jean Herbert | 225 | 0 | 225 | 262 | 262 | 0 |
| 8 | 0.00114476 | Peter Newmark | 271 | 0 | 271 | 360 | 360 | 0 |
| 9 | 9.87E−04 | Christine Nord | 166 | 0 | 166 | 212 | 212 | 0 |
| 10 | 9.34E−04 | Michael Halliday | 204 | 0 | 204 | 323 | 323 | 0 |
| 11 | 7.92E−04 | Mona Baker | 241 | 0 | 241 | 338 | 338 | 0 |
| 12 | 7.22E−04 | Herbert Grice | 83 | 0 | 83 | 101 | 101 | 0 |
| 13 | 7.11E−04 | Robin Setton | 271 | 0 | 271 | 376 | 376 | 0 |
| 14 | 6.80E−04 | Deirdre Wilson | 175 | 0 | 175 | 244 | 244 | 0 |
| 15 | 6.77E−04 | Dan Sperber | 174 | 0 | 174 | 207 | 207 | 0 |
| 16 | 6.71E−04 | Jef Verschueren | 58 | 0 | 58 | 79 | 79 | 0 |
| 17 | 6.21E−04 | David Carroll | 201 | 0 | 201 | 221 | 221 | 0 |
| 18 | 6.16E−04 | Gerald Miller | 103 | 0 | 103 | 114 | 114 | 0 |
| 19 | 6.05E−04 | John Anderson | 106 | 0 | 106 | 138 | 138 | 0 |
| 20 | 5.99E−04 | Miriam Shlesinger | 236 | 0 | 236 | 356 | 356 | 0 |
| 21 | 5.54E−04 | Barbara Moser-Mercer | 221 | 0 | 221 | 381 | 381 | 0 |
| 22 | 5.31E−04 | Roger Bell | 130 | 0 | 130 | 137 | 137 | 0 |
| 23 | 5.10E−04 | Wolfram Wilss | 132 | 0 | 132 | 150 | 150 | 0 |
| 24 | 4.97E−04 | Basil Hatim | 204 | 0 | 204 | 272 | 272 | 0 |
| 25 | 4.89E−04 | Ernst-August Gutt | 132 | 0 | 132 | 162 | 162 | 0 |
| 26 | 4.88E−04 | Jenny Thomas | 53 | 0 | 53 | 62 | 62 | 0 |
| 27 | 4.77E−04 | Christian Matthiessen | 71 | 0 | 71 | 80 | 80 | 0 |
| 28 | 4.75E−04 | Claire Kramsch | 76 | 0 | 76 | 82 | 82 | 0 |
| 29 | 4.49E−04 | Ian Mason | 183 | 0 | 183 | 231 | 231 | 0 |
| 30 | 4.49E−04 | Mark Shuttleworth | 85 | 0 | 85 | 92 | 92 | 0 |

**Table 3  The most influential works of Daniel Gile.**

| Cited works | PageRank scores |
|---|---|
| Basic concepts and models for interpreter and translator training (*Gile, 1995*) | 0.004854 |
| Conference interpreting as a cognitive management problem (*Gile, 1997*) | 2.15E−04 |
| Conference interpreting: current trends in research (*Gambier, Gile & Taylor, 1997*) | 1.27E−04 |
| Opening up in interpretation studies (*Gile, 1994*) | 1.19E−04 |
| Getting started in interpreting research (*Gile et al., 2001*) | 6.62E−05 |

**Table 4** The most influential works of Danica Seleskovitch.

| Cited works | PageRank scores |
| --- | --- |
| Pédagogie Raisonnée de l'Interprétation (*Seleskovitch & Lederer, 1989*) | 3.48E−04 |
| A systematic approach to teaching interpretation (*Seleskovitch & Lederer, 1995*) | 2.44E−04 |
| Language and cognition (*Seleskovitch, 1978b*) | 1.06E−04 |
| Interpréter pour traduire (*Seleskovitch & Lederer, 1984*) | 4.27E−05 |
| Translation of L'Interpréte dans les conférences internationale | 4.27E−05 |

**Table 5** The most influential works of Marianne Lederer.

| Cited works | PageRank scores |
| --- | --- |
| La traduction aujourd'hui: le modèle interprétatif (*Lederer, 1994*) | 6.18E−04 |
| Pédagogie Raisonnée de l'Interprétation (*Seleskovitch & Lederer, 1989*) | 3.48E−04 |
| A Systematic approach to teaching interpretation (*Seleskovitch & Lederer, 1995*) | 2.44E−04 |
| Simultaneous interpretation: units of meaning and other features (*Lederer, 1978*) | 1.31E−04 |
| Interpréter Pour Traduire (*Seleskovitch & Lederer, 1984*) | 4.27E−05 |

introducing methodologies from experimental psychology into interpreting research in the 1970s, but his work was heavily criticized by working interpreters for a lack of ecological validity, his manner of selecting research participants, and the methods he employed for evaluating their performances (*Gile, 1994*). This may explain his low PRA score, which indicates that the most influential Chinese scholars seldom cited his work—most citations of him were made by less influential researchers such as graduate students. The difficulty of accessing his publications in China may be another factor that contributes to his low score. A similar situation was observed for the rankings of Setton and Moser-Mercer: The former received the seventh most citations, as shown by his WDC score, but ranked only 13th in terms of influence. Having received his PhD from the Chinese University of Hong Kong in 1997 and subsequently taught in Taiwan and Shanghai, he remained active in CIS up to the time of his departure from the Shanghai International Studies University in 2011. While he published a number of texts on interpreting over the years, only his PhD dissertation, which was later published as a monograph, received much attention from the CIS community; the other papers he wrote received fewer than 15 incoming citations each (See Table 6). Moser-Mercer was cited by 221 Chinese colleagues, receiving a total of 381 citations, giving her a ranking of 12th in DC and 6th in WDC, but her position fell to 21st in the PRA. Examination of her works revealed that 39 of them received citations. The most cited was *Bridging the Gap: Empirical Research in Simultaneous Interpretation*—the first collected volume on empirical research into interpreting, on which she worked as a co-editor (see Table 7).

Working interpreters not engaged in research are still highly influential among Chinese academics. Roderick Jones is a case in point: only his monograph *Conference Interpreting Explained* (*Jones, 1998*), a manual that provides tips and techniques for honing

**Table 6 The most influential works of Robin Setton.**

| Cited work | PageRank scores |
| --- | --- |
| Simultaneous interpretation: a cognitive-pragmatic analysis (*Setton, 1999*) | 4.99E−04 |
| New demands on interpreting and the learning curve in interpreter training (*Setton, 2006*) | 5.67E−05 |
| Meaning assembly in simultaneous interpreting (*Setton, 1998*) | 5.45E−05 |
| Experiments in the application of discourse studies to interpreter training (*Setton, 1994*) | 5.45E−05 |
| The Geneva (ETI) perceptive on interpretation research (*Setton & Moser-Mercer, 2000*) | 4.27E−05 |

**Table 7 The most influential works of Barbara Moser-Mercer.**

| Cited work | PageRank scores |
| --- | --- |
| Quality in interpreting: some methodological issues (*Moser-Mercer, 1996*) | 1.23E−04 |
| Bridging the gap: empirical research in simultaneous interpretation (*Lambert & Moser-Mercer, 1994*) | 9.75E−05 |
| Beyond curiosity: can interpreting research meet the challenge (*Moser-Mercer, 1997a*) | 8.98E−05 |
| Simultaneous interpretation: a hypothetical model and its practical application | 8.83E−05 |
| Process models in simultaneous interpretation (*Moser-Mercer, 1997b*) | 7.55E−05 |

interpreting skills, received citations. However, his PRA ranking was 6th, immediately after Franz Pöchhacker. The professional's intuitive understanding of working interpreters is sometimes deeper and more comprehensive than that gained by collecting and analyzing empirical data, which might explain why he is highly cited by influential Chinese scholars. In addition, many may find his writing easy to relate to and his suggestions simple to follow in interpreting practice, contributing to his wide popularity in the CIS community.

A few translation scholars, such as Eugene Nida, Peter Newmark and Mona Baker, were also reasonably influential among Chinese academics. The first is noted for his Dynamic Equivalence Theory in translation, and works of his such as *Language, Culture, and Translating* (*Nida, 1993*), *Dynamic Equivalence in Translating* (*Nida, 1995*), and *Language and Culture: Context in Translating* (*Nida, 2001*) were widely cited by Chinese colleagues. This contributed to his PRA ranking of 4th among all Western scholars, behind Seleskovitch. He ranked 5th in DC, having been cited by 467 Chinese scholars, and received the same ranking for WDC with 730 citations. Though Pöchhacker outranked him in both DC and WDC, Nida's PRA score seems to suggest that he has a higher research impact than Pöchhacker in the CIS community. As mentioned earlier, the reason for the discrepancy between their DC/WDC and PRA scores is that Nida received more citations from influential CIS scholars than Pöchhacker, whereas the latter is more popular among low-ranking CIS scholars.

In addition, Western scholars in linguistics, sociology, cognitive science and psychology played an appreciable role in CIS research. For example, Lyle Bachman's research on language testing was often cited in work on the assessment of interpreting competence. Dan Sperber, a sociologist and cognitive scientist, developed the Relevance Theory in collaboration with Deidre Wilson, a psychologist by training. This theory has been used

**Table 8  Top 30 Chinese scholars in CIS by PRA.**

| No. | PageRank | Author's name | In-degree | Out-degree | Degree | Weighted degree | Weighted in-degree | Weighted out-degree |
|---|---|---|---|---|---|---|---|---|
| 1 | 0.00243402 | Mei Deming | 474 | 8 | 482 | 594 | 585 | 9 |
| 2 | 0.00213418 | Cai Xiaohong | 407 | 8 | 415 | 582 | 573 | 9 |
| 3 | 0.00137312 | Mu Lei | 125 | 9 | 134 | 160 | 148 | 12 |
| 4 | 0.00119857 | Bao Gang | 543 | 0 | 543 | 592 | 592 | 0 |
| 5 | 0.00119057 | Chen Jing | 110 | 183 | 293 | 388 | 147 | 241 |
| 6 | 0.00114121 | Li Yuqing | 10 | 1 | 11 | 12 | 11 | 1 |
| 7 | 0.0011069 | Zhong Weihe | 315 | 19 | 334 | 478 | 443 | 35 |
| 8 | 0.00106898 | Liu Heping | 513 | 0 | 513 | 843 | 843 | 0 |
| 9 | 9.33E−04 | Zhang Weiwei | 326 | 0 | 326 | 328 | 328 | 0 |
| 10 | 8.91E−04 | Liu Miqing | 415 | 0 | 415 | 498 | 498 | 0 |
| 11 | 6.69E−04 | Wu Bing | 102 | 0 | 102 | 107 | 107 | 0 |
| 12 | 6.59E−04 | Zhong Shukong | 305 | 0 | 305 | 311 | 311 | 0 |
| 13 | 6.11E−04 | Yang Chengshu | 195 | 30 | 225 | 283 | 247 | 36 |
| 14 | 6.00E−04 | Hu Gengshen | 159 | 13 | 172 | 250 | 237 | 13 |
| 15 | 5.96E−04 | Ke Keer | 37 | 0 | 37 | 37 | 37 | 0 |
| 16 | 4.91E−04 | Zhang Jiliang | 99 | 164 | 263 | 524 | 123 | 401 |
| 17 | 4.26E−04 | Zhuang Enping | 52 | 2 | 54 | 60 | 58 | 2 |
| 18 | 4.26E−04 | Sheng Qian | 93 | 6 | 99 | 107 | 101 | 6 |
| 19 | 4.00E−04 | Liu Minhua | 105 | 87 | 192 | 249 | 116 | 133 |
| 20 | 3.93E−04 | Li Yueran | 53 | 0 | 53 | 60 | 60 | 0 |
| 21 | 3.61E−04 | Xiao Xiaoyan | 178 | 14 | 192 | 219 | 199 | 20 |
| 22 | 3.56E−04 | Hu Qingping | 7 | 3 | 10 | 10 | 7 | 3 |
| 23 | 3.51E−04 | Zhang Junting | 10 | 0 | 10 | 11 | 11 | 0 |
| 24 | 3.35E−04 | Li Changshuan | 85 | 12 | 97 | 118 | 105 | 13 |
| 25 | 3.25E−04 | Yi Honggen | 32 | 0 | 32 | 33 | 33 | 0 |
| 26 | 3.19E−04 | Mu Yuanyuan | 22 | 0 | 22 | 22 | 22 | 0 |
| 27 | 3.19E−04 | Pan Jun | 22 | 0 | 22 | 22 | 22 | 0 |
| 28 | 3.17E−04 | Wang Dawei | 109 | 3 | 112 | 128 | 125 | 3 |
| 29 | 3.01E−04 | Chen Siqing | 7 | 0 | 7 | 7 | 7 | 0 |
| 30 | 2.88E−04 | Li Nanqiu | 82 | 2 | 84 | 87 | 85 | 2 |

by numerous Chinese scholars to shed light on the processes of listening comprehension, note-taking and language production in interpreting.

The data reveal the following to be the top 30 most influential Chinese scholars (see Table 8).

In comparison with their Western colleagues, the composition of the top-ranking Chinese scholars in PRA was quite homogenous. Of the top 30, no fewer than 25 (all but Mei Deming, Liu Miqing, Wu Bing, Hu Gengshen and Zhuang Enping) are interpreting scholars.

Cai Xiaohong ranked 5th in DC and 6th in WDC, but had the second highest PRA score of all, indicating that she had a very large research impact among her fellows. In 2000, she defended her doctoral dissertation, in which she studied the development of competence

**Table 9 The most influential works of Cai Xiaohong.**

| Cited work (original Chinese title) | Englished title | PageRank scores |
| --- | --- | --- |
| 以跨学科的视野拓展口译研究 | Interpretation study with an interdisciplinary perspective (*Cai, 2001a*) | 4.32E−04 |
| 交替传译过程及能力发展 | The process of consecutive interpreting and skills development (*Cai, 2001b*) | 1.77E−04 |
| 论口译质量评估的信息单位 | Assessing interpreting quality: an approach based on units of meaning (*Cai, 2003*) | 1.35E−04 |
| 口译研究新探 | An exploration of interpreting research (*Cai, 2002*) | 7.08E−05 |
| 口译评估 | Interpreting assessment (*Cai, 2007*) | 4.26E−05 |

**Table 10 The most influential works of Bao Gang.**

| Cited work (original Chinese title) | Englished title | PageRank scores |
| --- | --- | --- |
| 口译理论概述 | An overview of interpreting theories (*Bao, 1998a*) | 0.00266572 |
| 高校口译训练的方法 | Preparatory training for undergraduate interpreting students (*Bao, 1992*) | 1.51E−04 |
| 口译程序中的思维理解 | Reasoning and comprehension in the interpreting process (*Bao, 1999*) | 9.00E−05 |
| 口译程序中的语义问题 | Semantic issues in the interpreting process (*Bao, 1998b*) | 6.85E−05 |
| 译前准备术语强记的方法论 | Methods for memorizing terms before an assignment (*Bao, 1996*) | 6.81E−05 |

in consecutive interpreting by conducting an experiment with 12 participants of different skill levels. Since then, she has published a number of monographs and papers, of which the top five most frequently cited are shown in Table 9. The present research revealed that as was the case for Moser-Mercer, Cai's most cited work was a collective volume which she co-edited. All the other top five works were research papers she has published over the years.

Bao Gang ranked 1st in DC, having been cited by 543 people and receiving 592 citations, the majority of them on his monograph *An Overview of Interpreting Theories*, but his ranking dropped to 4th in PRA. A possible explanation for this is that sadly he passed away in 1999 at the age of 42: had he lived, he would surely have contributed a great deal more to CIS. In Table 10, which lists his top five most cited works, it is interesting to observe that the citation distribution for his works is lopsided: the overwhelming majority of scholars cite his aforementioned work, an introduction to the leading Western theories prevalent in the 1990s. A possible explanation for this highly skewed distribution is, once again, his untimely death.

Liu Heping ranked 2nd in both DC and WDC, receiving a total of 843 citations from 513 scholars, but her ranking dropped to 8th in PRA. Her most cited works are shown in Table 11. Liu's incoming citations seem much more balanced than those of the previously mentioned CIS scholars. Though her monograph *Interpreting Techniques: Scientific Thinking and Reasoning* received the majority of incoming citations, her papers, which occupied the rest of the top five spots, were also cited by numerous CIS scholars. Liu obtained her PhD in Translation Studies from the University of Paris III, and is an

**Table 11  The most influential works of Liu Heping.**

| Cited work (original Chinese title) | Englished title | PageRank scores |
| --- | --- | --- |
| 对口译教学统一纲要的理论思考 | A few thoughts on standardized interpreter teaching plans (*Liu, 2002*) | 2.86E−04 |
| 口译理论研究成果与趋势浅析 | Trends in interpreting research (*Liu, 2005*) | 2.70E−04 |
| 翻译的动态研究与口译训练 | Research in translator and interpreter training (*Liu, 1999a*) | 2.57E−04 |
| 口译技巧：思维科学与口译推理教学法 | Interpreting techniques: scientific thinking and reasoning (*Liu, 2001a*) | 1.15E−04 |
| 口译理论与教学现状研究及展望 | Interpreting theories and teaching of today and tomorrow (*Liu, 2001b*) | 4.45E−05 |

active member of the CIS community, regularly appearing as a keynote speaker at various conferences. A former student of Seleskovitch's, she continues to advocate the Interpretive Theory of Translation in her research.

As is the case for their Western colleagues, the opinions of Chinese practicing interpreters are also highly valued by CIS scholars. Zhang Weiwei, who ranked 9th in PRA, once served as a staff interpreter at the United Nations' duty station in Geneva: he was cited almost exclusively for his handbook *English-Chinese Simultaneous Interpreting* (*Zhang, 1999*), which highlights the use of segmentation and syntactic linearity with numerous practical examples. In addition, his high rankings in DC (9th) and WDC (12th) suggest that he was equally popular with scholars both influential and less so.

Zhang Wei's rankings in PRA, DC and WDC merited some investigation. He topped the list in WDC, his ranking having been significantly boosted by his weighted out-degree centrality (WODC)—he made a total of 854 citations of other scholar's works in all his papers and in his doctoral dissertation, far exceeding the second highest in WODC, Ren Wen, whose total was 557. However, in DC Zhang slipped to 6th position, and in PRA to 44th. It should also be noted that he was the most productive CIS author, with 28 papers recorded in the data-set. The discrepancy between his influence score and research productivity might be explained by his being a relative newcomer to CIS, having received his PhD in 2007—it is common that the older a work is, the larger its readership and the more it will have been cited. Another plausible reason for the discrepancy is that his doctoral dissertation, in which he investigated the relationship between simultaneous interpreting and working memory, was highly technical and used a combination of experimental, observational and questionnaire-based research methods; the unfamiliar techniques he used may have deterred other CIS authors from adopting his work. Furthermore, the research carried out for Question 3 of the present study revealed that authors writing about cognitive issues were slightly more likely to end up in the bottom PageRank influence group than those writing on other topics.

There are also a number of scholars who, though trained and specializing in other disciplines, occasionally venture into the field of interpreting research, namely Mei Deming, Hu Gengshen, Liu Miqing, Wu Bing and Zhuang Enping. Mei topped the Chinese scholars' PRA rankings. He is among the most prolific MA thesis advisors in China, and was involved in launching the Shanghai Interpreting Exams. A PhD graduate in linguistics and

**Table 12 Statistically significant themes in multinomial regression for PageRank.**

| Influence group | Theme group | Change in likelihood of belonging to an influence group (%) | p-value |
|---|---|---|---|
| Medium (>=20th, <85th percentile) | Cognitive | −0.4 | 0.068 |
| | Socio-Cultural | −0.6 | 0.009 |
| High (>=85th percentile) | Miscellaneous | 0.7 | 0.029 |

rhetoric from the University of Indiana, Mei has taught numerous classes on linguistics, public speaking and movie appreciation. In the data-set there were no research papers on interpreting written by him alone, though he co-authored a few with his doctoral students. In addition to all the above, Mei was the editor-in-chief for *An Advanced Course in Interpreting* (*Mei, 1996*), which was first published in 1996 (new editions in 2000, 2005 and 2011), and for *An Intermediate Course in Interpreting* (*Mei, 1998*), first published in 1998 (new editions in 2003, 2008 and 2010). These are the mandatory test preparation books for the Shanghai Interpreting Exams, and other Chinese authors' citing of them contributed to his becoming one of the most influential CIS authors in the country. The case of Wu Bing follows the same lines: she is mainly cited for her textbook on Chinese-to-English interpreting. Neither author has published any empirical studies on the subject. Zhuang Enping specializes in cross-cultural communication, but he has also written articles on the principles of interpreting and how the differences between Eastern and Western styles of communication affect it. His high degree of influence among Chinese scholars (17th in PRA) indicates that a good number of researchers draw inspiration from Communication Theory.

Unlike the situation with the top 30 most influential Western scholars, the top 30 Chinese list had only one Chinese scholar whose research was completely unrelated to interpreting. Chen Siping, who ranked 29th in PRA, focused on the application of the Relevance Theory to reading comprehension; her high ranking would seem to indicate that influential Chinese scholars frequently used her work as the theoretical underpinnings for their research.

## Research topics and academic influence
### *Summary of significant findings*

As stated in the methodology section, inferences drawn from linear regressions were somewhat unsatisfactory: F-tests proved only the linear model for Out-Degree to be statistically significant.

As discussed then, we consequently turned to multinomial regressions for three influence groups—low, middle and high—for each of the network measures: PageRank, In-Degree, Out-Degree, and EigenVector Centrality. We delimited the group divisions by determining the cutoff points that led to the most statistically significant regression models.

For each measure and each influence group we report below the themes that had statistically significant coefficients (see Tables 12–14).

**Table 13  Statistically significant themes in multinomial regression for in-degree.**

| Influence group | Theme group | Change in likelihood of belonging to an influence group (%) | p-value |
| --- | --- | --- | --- |
| Medium (>=60th percentile, <95th percentile) | Language | 0.6 | 0.011 |
| High (>=95th percentile) | Socio-Cultural | 1.11 | 0.006 |

**Table 14  Statistically significant themes in multinomial regression for out-degree.**

| Influence group | Theme group | Change in likelihood of belonging to an influence group (%) | p-value |
| --- | --- | --- | --- |
| Low (=0th percentile) | Language | −0.4 | 0.094 |
| Medium (>0th percentile, <80th percentile) | Professional | 0.8 | 0.036 |
| High (>=80th percentile) | Cognitive | 0.9 | 0.000 |

The change in likelihood column shows that authors who had 1% more publications in the listed theme were $x$% more or less likely to be in that influence group for that influence measure. Taking PageRank (see Table 12) as an example: when authors wrote 1% more publications that fell into the Cognitive theme category, they were 0.4% less likely to be part of the Medium-ranking influence group. In other words, an author with no Cognitive keywords in 100% of his publications would be 40% more likely to have medium PageRank than an author all of whose keywords were Cognitive.

Although the linear regressions had low statistical power, the trends they predicted coincide with the findings of the multinomial regressions (see sections 'Linear regression: a first approach to modeling the relationship between network influence and memes' and 'Multinomial logistic regression: procedure for stratifying authors into high-, middle- and low-ranking groups' below for numerical results). In particular, both Cognitive and Language keywords increased an author's Out-Degree measure. The multinomial regression analysis clarified how these keywords increased an author's Out-Degree measure. In particular, Cognitive keywords increased an author's likelihood of having high Out-Degree, while Language keywords decreased the likelihood of his having low Out-Degree.

The Lasso multinomial regressions for keywords (see section 'Regularized multinomial regression for predicting influence by keywords') also supported the earlier multinomial regression results for theme category analysis. The consistency of these analyses is supported by the fact that the majority of the keywords, and the theme categories that these keywords belong to, share the same correlation sign (either positive or negative) as the influence measure group of the authors who wrote them. To illustrate this with an example, the keyword "Theory" was positively correlated with an author's placement in the High-ranking group of PageRank, and the theme category of this keyword (Miscellaneous) was also found to be positively correlated with an author's likelihood of belonging to that same High-ranking group. It should also be acknowledged that the Lasso regression analysis failed to detect a couple of findings from the theme category analysis. For example,

**Table 15** *F-test for different network influence measures.*

|  | PageRank | In-degree | Out-degree | Eigenvector centrality |
|---|---|---|---|---|
| *P*-value | 0.348 | 0.183 | 0.01 | 0.176 |

the positive relationship between Social-cultural keywords and the High In-Degree influence group, and the negative relationship between Language keywords and the Low Out-Degree group, were not picked up by the Lasso model. It is likely that those keywords were not significant on their own, but collectively they contributed to boosting an author into a certain influence group.

In sum, from the findings above it was found that the most influential authors are those who write about Social-cultural and non-mainstream topics. In particular, authors whose papers cover Theory are more likely to be placed in the High influence group than those who do not. In addition, the analysis revealed that those writing about Cognitive issues, especially Sensory Memory, are more likely to include a significant number of bibliographic references.

### Linear regression: a first approach to modeling the relationship between network influence and memes

For the preliminary analysis four standard linear regressions of each of the metrics were performed using the theme profiles as predictors. The fact that each theme profile was normalized might result in prolific and non-prolific authors having very similar profiles. Therefore, the number of papers published was added as an extra predictor. This was done to separate the effects of an author's theme profile from his or her overall frequency of publishing. Table 15 shows the F-test *p*-values for each model.

The only regression model which turned out to be significant was Out-Degree. In general terms, this means that the topics an author writes about are a good predictor of the number of outgoing citations in his papers. For example, when people write about cognitive issues, the coefficient of this theme (see Table 16 below) suggests that they tend to have higher Out-Degree scores, meaning that they cite more papers. In sum, the coefficients in Table 16 essentially indicate that the technical nature of an author's papers can be a reliable predictor of his ranking in Out-Degree measure. A possible explanation for this is that more technical papers require the researcher to conduct extensive literature review in order to ease the reader into his own research. This often translates into a large number of citations in his or her papers.

The coefficients of themes in Table 16 represent the changes that would occur in Out-Degree if that particular theme proportion were increased by 1%. For example, a 1% shift in an author's theme profile to Cognitive would result in a 0.08381 increase in Out-Degree. A similar 1% shift to Language would result in a 0.06429 increase. It should be noted, however, that the coefficient for the number of papers represents the amount of Out-Degree shifted as a result of increasing the number of papers by 1. More specifically, the coefficient for the number of papers indicates that having one more paper published might corresponds to a 0.59 increase in predicted Out-Degree for that particular author.

**Table 16  Linear regression output of themes on out-degree centrality measure.**

| Coefficients | Estimate | Std. error | Pr(>|t|) |
|---|---|---|---|
| (Intercept) | 8.85850 | 1.17883 | 8.3e−14 |
| **No. of papers** | **0.59477** | **0.36048** | **0.0991** |
| **Cognitive** | **0.08381** | **0.02687** | **0.0018** |
| **Language** | **0.06429** | **0.02656** | **0.0156** |
| Miscellaneous | 0.04645 | 0.03449 | 0.1783 |
| Professional | 0.04377 | 0.04183 | 0.2955 |
| Socio-cultural | −0.00565 | 0.02865 | 0.8436 |
| Multiple R-squared | 0.0078 | | |
| Adjusted R-squared | 0.00501 | | |
| F-statistic: *p*-value | **0.0103** | | |

It should be noted that a coefficient for Training does not appear in the regression summary. This is due to the profile normalization discussed earlier. If an author's theme profile is known for any five terms, their score for the sixth theme can be precisely inferred by *1—sum (scores for 5 themes)*. Hence, when an author's theme profile is parameterized to five dimensions, as is necessary for computational reasons, it contains the same information as a full 6-theme profile. Each theme profile group was treated as a numeric variable so that the effect of changing the relative proportion of that theme would be clearly visible. Any positive coefficient in the regression means that trading off Training keywords for keywords in that particular theme profile results in an increased Out-Degree.

Relative effects can be revealed by using algebraic manipulations of the coefficient estimates and adjusting for Training. For example, consider the impact of trading a percentage of the Cognitive theme for the Language theme. The model loses 0.08 Out-Degree (that is, 0.08 outgoing citations, or 8% of one outgoing citation) by trading 1% of the Cognitive theme for 1% of the Training theme; it gains 0.06 Out-Degree by trading Training for Language. The following formula shows how these values are computed:

$$Effect(-1\ cognitive, +1\ language) = Effect(-1\ cognitive, +1\ training)$$
$$+ Effect(-1\ training, +1\ language) = -0.08 + 0.06 = -0.02.$$

Trading off 1% of Cognitive theme for 1% of Language theme lowers Out-Degree by 0.02. If this shift in themes were to be repeated 50 times, the model predicts that the paper would have its Out-Degree score reduced by 1 (that is, one fewer outgoing citation).

The analysis thus reveals that using more Cognitive or Language themes in papers is a predictor of having higher Out-Degree counts, i.e., authors writing on those subjects generate more references. The data also suggest that having larger numbers of papers published is correlated, although only marginally, with higher Out-Degree; an interesting finding, in that one would expect writing more papers to lead to an inevitable increase in the total number of outgoing citations. There are two possible explanations for this marginal effect. Firstly, the fact that 30% of the CIS papers in the data-set do not have any references suggests that a large proportion of authors do not necessarily have a high

**Table 17  Gini coefficients for four network influence measures.**

| Network influence | PageRank | In-degree | Out-degree | Eigenvector |
|---|---|---|---|---|
| Gini coefficient | 0.28 | 0.83 | 0.81 | 0.84 |

number of outgoing citations, even if they have written multiple publications. Secondly, it is possible that a few highly prolific authors generate only small numbers of outgoing citations, which may have an effect on the correlation between the number of publications and the Out-Degree measure.

It should be noted that the multiple and adjusted R-squared values in Table 16 are rather low—less than 1% of the variance in the data is explained by the model, even though the *p*-values for several predictors are very small. This means that the results of standard linear regression are not very promising: while the trend detected is significant, the model still cannot explain the data very well. Huge disparities between authors' network influence measures were a likely cause for this inadequacy. A large discrepancy or inequality in a response variable makes a linear regression inadequate because estimated effects are highly influenced by a few extreme values, and hence effects which pertain to the rest of the population are subsumed. In the following section we examine this inadequacy further, and describe mathematically the amount of network measure inequality through an analysis of Gini coefficients.

### Gini coefficient: measuring disparities in each network measure across authors

The Gini coefficient has typically been used to calculate income inequality in populations, by converting the cumulative distribution of wages into a single number. In such cases, a Gini value of 0 corresponds to complete income equality, i.e., every individual is earning exactly the same wage, whereas a value of 1 corresponds to complete inequality, i.e., one individual is in receipt of all the wealth. For the present study, the procedure was applied to evaluate the distribution of our four network influence measures; the results are summarized in Table 17. Once the procedure was completed, we matched these Gini coefficients to publicly reported wage distributions of countries, allowing us to form analogies which demonstrate the amount of inequality in network measures and hence the inadequacy of linear regression.

The analysis revealed that PRA has the most equitable distribution of ranks, roughly equal to the amount of wage inequality in Belgium.[9] On the other hand, the other ranks are much less balanced, with larger inequality than the wage inequality in any country measured by the World Bank—the one with the largest recorded Gini coefficient is the Seychelles, with 0.658 in 2007. Despite PRA's having a relatively equitable distribution of ranks, the PRA scores of CIS authors were still not good enough for linear regression analysis: 200 authors (about 10% of the CIS total) control 30% of the scores, though the remaining 70% are almost perfectly equally distributed among the remaining 1,950. This means that we are missing inference on the remaining 1,950 authors because linear regression of the PageRank score focuses on the largest difference: the one between the

[9] All references to world income inequality calculations are taken from the World Bank's Gini index: http://data.worldbank.org/indicator/SI.POV.GINI/.

1,950 authors and the top 200. This calls for a better approach to evaluating the impact of an author's theme profile on his or her rankings.

### Multinomial logistic regression: procedure for stratifying authors into high-, middle- and low-ranking groups

Though linear regression was the first choice—and most straightforward—method of explaining authors' levels of influence based on their theme profiles and numbers of publications, as we have seen it turned out to be lacking when it came to explaining their influence within networks. An alternative approach was to classify the authors into three influence groups—low, middle and high—using multinomial regression.

The three groups can be defined in different ways depending on which cutoffs are used to separate them. Only two parameters are needed to define the three groups: one cutoff value to separate the low and middle groups and another to separate the middle and the high. N.B.: By knowing these two values we can know both the length and the midpoint value of the middle influence group.

The multinomial models represent the probability of an author's being in each of the three groups, given an author's theme profile and number of papers written. The coefficients of each of the multinomial models were determined by fitting each model to the data, where the response variable is now an indicator of each author's allocation to the three groups corresponding to each model. The coefficients depended on how the three groups were divided, and each pair of cutoffs was associated with a different multinomial model. In Figs. 6 and 7, each square represents a model and is thus associated with a certain definition of the three groups of authors, or equivalent to certain values of the two cutoffs.

Figure 6 shows the results of fitting each of the 209 group divisions for PageRank, and calculating a $p$-value for the null hypothesis of no significant relation between theme profile and group membership. Since we expect about 21 groups to have $p$-values less than 0.1 completely by chance, it is not instructive to report all groups with low $p$-values as indicative of significance. Many of these low $p$-values will not relate to significant effects and would confuse results.

Instead, the quality of a group of models was evaluated by a so-called False Discovery Rate (FDR) analysis. In the present study, the FDR of a group of models is the expected proportion of models that are not good, i.e., are not statistically significant ones, therefore the lower the FDR the better the group of models. The quality of each model was measured by a quantity called $q$-value. Simply put, to build a group of models with an FDR lower than a certain threshold value, only models with a $q$-value lower than that value can be included. The results of this FDR analysis for PageRank are summarized in Fig. 7.

Figure 7 shows that the FDR-based approach makes it easy to identify groups of good models: there are three stripes of squares associated with a small FDR. The furthest (red) stripe to the right corresponds to the models whose group divisions yield the most statistically significant results, i.e., which best explain the connection between an author's theme profile and his influence.

The red squares correspond to $q$-values lower than 0.05, so the group of models corresponding to the red squares has an FDR below 5%. In other words, no more than

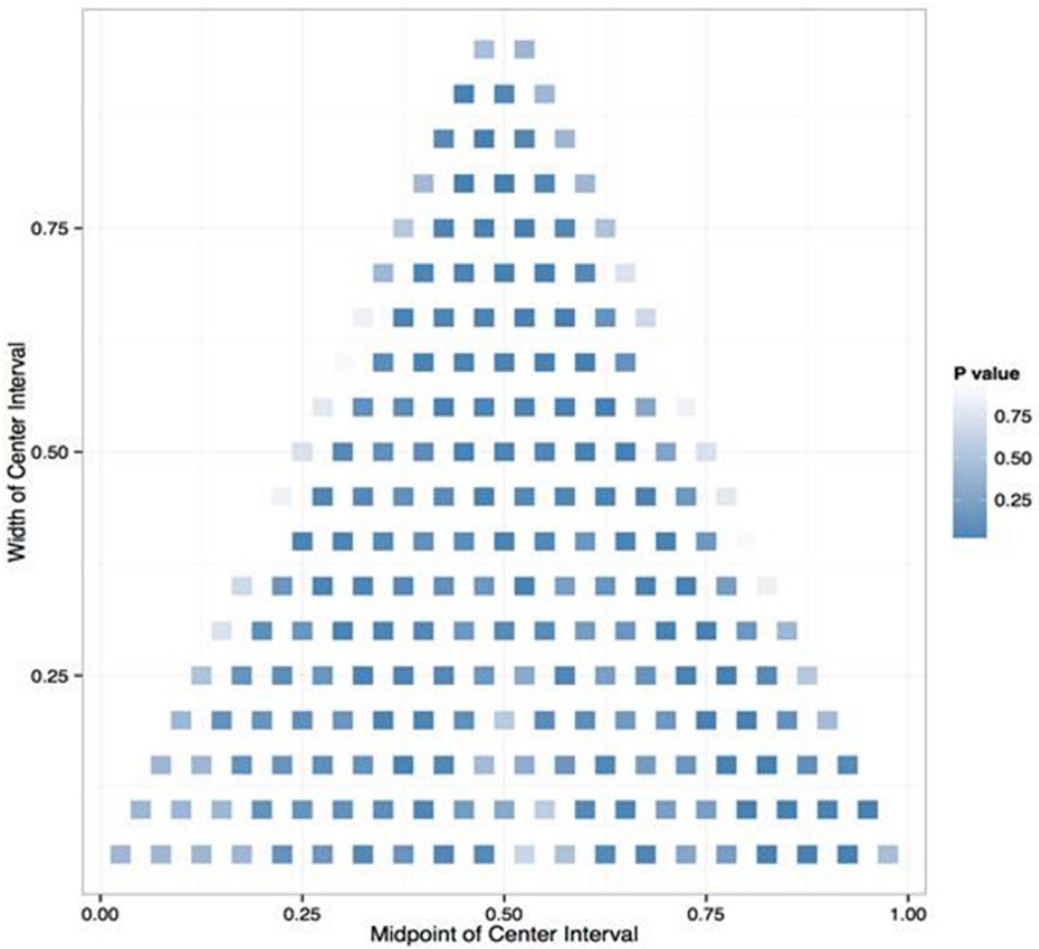

**Figure 6** *P*-value of the deviance test for a significance multinomial regression over all 209 possible PageRank group divisions. The divisions are parameterized by the midpoint of the Middle Ranking group on the *x* axis and by its width on the *y* axis. Darker squares denote more significant regressions and should be regarded as indicating that these divisions are significantly influenced by theme profile.

5% of these models are expected to be non-significant. Similarly, since the orange squares correspond to *q*-values above 0.05 and below 0.10, all groups of models, whether they correspond only to orange squares or to orange and red ones, have an FDR below 10%. The same reasoning can be repeated for each color of square and its associated *q*-values. In Fig. 5, the three diagonal stripes of red and orange squares correspond to the best and relatively good models described in this paragraph.

Once groups of relatively good models (each defined by a pair of cutoffs) had been determined, the selection of a 'stand-out' model, i.e., one with an exceptionally good pair of cutoffs, was still required. A clustering procedure known as k-medoids, by which similar pairs of cutoffs are divided into different groups, was employed at this point (*Kaufman & Rousseeuw, 1987*). The chosen divisions corresponded to the most central point in the cluster with the smallest FDR (i.e., the cluster with the best statistical significance). The best pair of cutoffs for PageRank are the 20th percentile for low-ranking authors and the 85th for high-ranking ones (see Fig. 7).

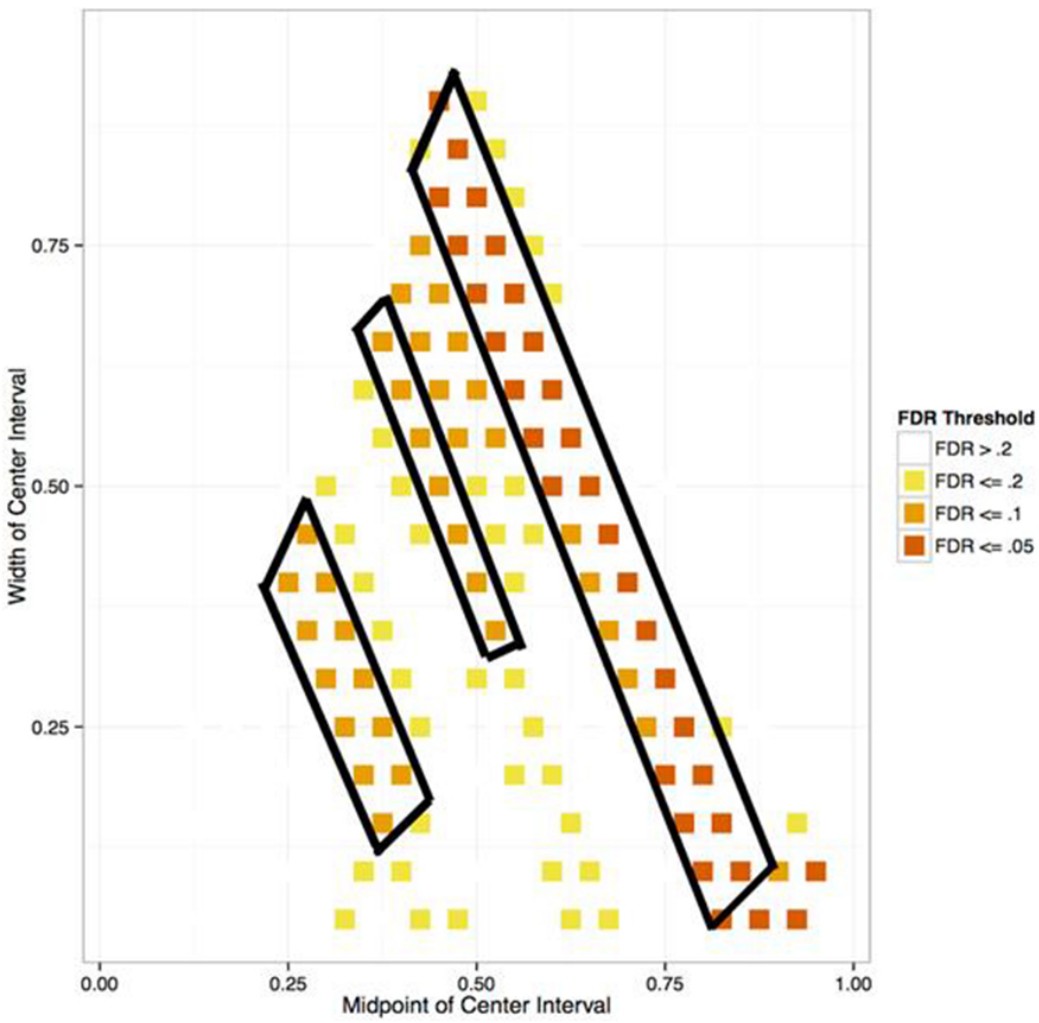

**Figure 7** *q*-**values for PageRank influences.** The *q*-value is a False Discovery Rate (FDR) analysis analogue of the *p*-value. It is the minimal FDR at which the test result is considered significant. The best pair of cutoffs for PageRank are the 20th percentile for low-ranking authors and the 85th for high-ranking ones.

Similar analysis revealed the optimal cutoffs for In-Degree to be [0.6, 0.95] and for Out-Degree [0.6, 0.8]. Theme profiles best explain the differences between the bottom 60% of Out-Degrees (less than two incoming citations), the top 5% (more than 33 incoming citations), and the middle group (those between 60 and 95%). Similarly, they best explain the differences between the bottom 60% of In-Degrees (0 outgoing citations), the top 20% of Out-Degrees (more than 24 outgoing citations), and the middle group. Note that for Out-Degree we can choose any cutoff below 64% as that percentage of authors had 0 Out-Degree citations. Hence, any such quantile choice would result in absolute cutoffs of 0 and 24 outgoing citations. For EigenVector, no cutoffs were found that had a chance of less than 20% of being non-null hypotheses—in fact, no significant *p*-values at all came to light. As a result, the predictors worked better with PageRank than EigenVector for measuring author influences in the CIS network; for this reason, the latter measurement

**Table 18  Optimal cutoff quantiles and their measure values.**

|  | Low cutoff (percentile) | Low cutoff (Val) | High cutoff (percentile) | High cutoff (Val) |
|---|---|---|---|---|
| PageRank | 20 | 0.0000659 | 85 | 0.0000877 |
| In-degree | 60 | 2 | 95 | 33 |
| Out-degree | 60 | 0 | 80 | 24 |

was not adopted for the following analysis. Table 18 shows optimal cutoff quantiles and their corresponding measure values.

The output shown in Tables 19, 20 and 21 below summarizes the results of multinomial regression of theme profiles on the three network measures. The coefficients are symmetric relative odds on the log scale.

We present all of our results on the relative odds scale, which can be found from the transformation $\exp(\mathrm{coef})$. This makes it easier to compare the relative effect of different theme profile variables in different subgroups. For example, a coefficient of $-0.004$ for cognitive in the middle group for PageRank means that authors who have 1% less cognitive articles have on average $1 - \exp(-0.004) = 0.006$ percent smaller odds. It is straightforward to transform odds into raw percentages by taking into account the intercept coefficient and an author's theme profile. Consider an author who has 100% cognitive articles: They would have $\exp(0.988 - 100 * .004) = 1.8$ to 1 odds, or about a $1.8/(1 + 1.8) = 0.62$ chance of being in the middle group.

For PageRank, it was found that trading off 1% of Cognitive for 1% of any other paper gave us $\exp(-0.004) = 0.996$ lower odds, or roughly a 0.4% lower likelihood of being in the middle group of PageRanks. The same trade-off for Socio-cultural gave us $\exp(-0.006) = 0.994$ lower odds, or roughly a 0.6% lower likelihood. The only positive significant relationship between theme profile and placement in the high PageRank group came from trading for Miscellaneous papers. Every percentage point increase in Miscellaneous papers gave roughly 1.007 greater odds of being in the top 15% of PageRanks. This is attributable to the fact that some highly influential papers were classified as Miscellaneous. Earlier research (*Xu, 2014*; *Xu, 2015*) indicated that the Miscellaneous theme accounts for only 9% of the total themes mentioned in MA theses, 0% in doctoral dissertations and 5.1% in research papers, meaning that non-mainstream topics have been explored by few CIS researchers, whereas numerous papers on Cognitive, Language, Professional and Socio-cultural issues have competed for attention. It appears to be much easier for those authors with papers on Miscellaneous subjects to get noticed and be cited by other researchers.

Another way of describing a 0 coefficient for a theme profile variable for a certain group is as follows: when one percentage point of this variable is exchanged for one of the Training theme, there is no change in the probability of an author's belonging to the group under consideration. For example, when an author trades 1% of Cognitive or Language for 1% of Training, the probability of his belonging to the high-ranking PageRank group is not affected at all, but that probability is reduced by 0.007 when he trades 1% of Miscellaneous for 1% Training theme.

**Table 19  PageRank multinomial regression.**

|  | coef | se | p values |
|---|---|---|---|
| **Low (<20th percentile)** | | | |
| **Intercept** | **−0.270** | **0.110** | **0.014** |
| No. of papers | 0.000 | 0.031 | 1.000 |
| Cognitive | 0.002 | 0.002 | 0.508 |
| Language | 0.002 | 0.002 | 0.432 |
| Miscellaneous | 0.000 | 0.003 | 1.000 |
| Professional | 0.000 | 0.004 | 0.978 |
| Socio-cultural | 0.000 | 0.003 | 1.000 |
| **Medium (>=20th, <85th percentile)** | | | |
| **Intercept** | **0.988** | **0.095** | **0.000** |
| No. of papers | −0.036 | 0.028 | 0.207 |
| **Cognitive** | **−0.004** | **0.002** | **0.068** |
| Language | −0.001 | 0.002 | 0.763 |
| Miscellaneous | −0.003 | 0.003 | 0.276 |
| Professional | −0.002 | 0.003 | 0.569 |
| **Socio-cultural** | **−0.006** | **0.002** | **0.009** |
| **High (>=85th percentile)** | | | |
| **Intercept** | **−0.719** | **0.129** | **0.000** |
| No. of papers | 0.003 | 0.035 | 0.936 |
| Cognitive | 0.000 | 0.003 | 1.000 |
| Language | 0.000 | 0.003 | 1.000 |
| **Miscellaneous** | **0.007** | **0.003** | **0.029** |
| Professional | 0.000 | 0.005 | 1.000 |
| Socio-cultural | 0.002 | 0.003 | 0.512 |

The next item to be scrutinized was the regression on In-Degree. The In-Degree multinomial results showed that trading off any theme profile for Language papers gave 1.006 higher odds of being in the middle In-Degree group (greater than 2 but fewer than 33 incoming citations). It was also observed that if an author traded off any theme profile for the Socio-cultural themes, they would have 1.011 higher odds to be in the the high In-Degree group (having more than 33 incoming citations). This was the most significant effect of any theme on any group, suggesting that authors on Socio-cultural issues are easily identified and cited by their confreres. As was the case for the Miscellaneous theme, Socio-cultural issues received little—though slightly greater—attention from CIS researchers and authors: 10.7% for research papers, 11.1% for MA theses and 12.4% for doctoral dissertations. Socio-cultural issues play an important part in interpreting, which can all too easily be affected by factors such as contexts and hidden power relations between various actors in the dialogue (*Pym, Shlesinger & Jettmarová, 2006*).

Finally, the same procedure was applied for Out-Degree measure. Here it was observed that if authors wrote more Language papers, they were $\exp(−0.004) = 0.996$, or had 0.004 lower odds to end up generating the least number of citations (low rank group). It should be noted, however, that since the $p$-value in this case was 0.09, this finding is marginal

**Table 20  In-degree multinomial regression.**

|  | coef | se | *p* values |
|---|---|---|---|
| **Low (<60th percentile)** |  |  |  |
| **Intercept** | **1.237** | **0.097** | **0.000** |
| No. of papers | 0.000 | 0.028 | 1.000 |
| Cognitive | −0.001 | 0.002 | 0.677 |
| Language | 0.000 | 0.002 | 1.000 |
| Miscellaneous | −0.003 | 0.003 | 0.309 |
| Professional | −0.002 | 0.003 | 0.473 |
| Socio-cultural | 0.000 | 0.002 | 1.000 |
| **Medium (>=60th percentile, <95th percentile)** |  |  |  |
| **Intercept** | **0.194** | **0.101** | **0.054** |
| No. of papers | 0.041 | 0.029 | 0.153 |
| Cognitive | 0.000 | 0.002 | 1.000 |
| **Language** | **0.006** | **0.002** | **0.011** |
| Miscellaneous | 0.000 | 0.003 | 1.000 |
| Professional | 0.001 | 0.004 | 0.799 |
| Socio-cultural | −0.001 | 0.003 | 0.613 |
| **High (>=95th percentile)** |  |  |  |
| **Intercept** | **−1.431** | **0.238** | **0.000** |
| No. of papers | −0.100 | 0.100 | 0.318 |
| Cognitive | 0.002 | 0.005 | 0.739 |
| Language | −0.003 | 0.005 | 0.601 |
| Miscellaneous | 0.005 | 0.005 | 0.308 |
| Professional | 0.000 | 0.007 | 1.000 |
| **Socio-cultural** | **0.011** | **0.004** | **0.006** |

at best. In addition, an author had a 1.008 higher probability of having between 1 and 24 outgoing citations (middle rank group) by writing more Professional papers, 1.009 higher odds of at least 24 citations (high rank group) by writing more Cognitive papers, and 1.077 higher odds of at least 24 outgoing citations by writing more papers in general. These results were found generally to tally with those of the linear regression described in section 'Linear regression: a first approach to modeling the relationship between network influence and memes.'

### *Regularized multinomial regression for predicting influence by keywords*

A regularization technique called Lasso was run for multinomial regression (*Tibshirani, 1996*) with 10-fold cross-validation to approximate the optimal set of keywords which were truly significant. The Lasso works by applying a penalty to the absolute value of coefficients, providing a principled way to set the coefficients of non-significant keywords to 0. Any remaining keywords were considered to be significantly related to the network measure. For the dependent variables, the same optimal cutoffs were used as those found by the FDR analysis examined in Table 18. By this technique, keywords that were not relevant enough for the present analysis could be discarded.

**Table 21  Out-degree multinomial regression.**

| | coef | se | *p* values |
|---|---|---|---|
| **Low (=0th percentile)** | | | |
| **Intercept** | **1.014** | **0.096** | **0.000** |
| No. of papers | 0.000 | 0.029 | 1.000 |
| Cognitive | 0.000 | 0.002 | 1.000 |
| **Language** | **−0.004** | **0.002** | **0.094** |
| Miscellaneous | −0.004 | 0.003 | 0.180 |
| Professional | 0.000 | 0.003 | 1.000 |
| Socio-cultural | −0.002 | 0.002 | 0.372 |
| **Medium (>0th percentile, <80th percentile)** | | | |
| **Intercept** | **−0.469** | **0.130** | **0.000** |
| No. of papers | −0.029 | 0.046 | 0.528 |
| Cognitive | −0.001 | 0.003 | 0.837 |
| Language | 0.000 | 0.003 | 1.000 |
| Miscellaneous | 0.001 | 0.003 | 0.813 |
| **Professional** | **0.008** | **0.004** | **0.036** |
| Socio-cultural | 0.001 | 0.003 | 0.647 |
| **High (>=80th percentile)** | | | |
| **Intercept** | **−0.545** | **0.117** | **0.000** |
| **No. of papers** | **0.075** | **0.031** | **0.014** |
| **Cognitive** | **0.009** | **0.002** | **0.000** |
| Language | 0.002 | 0.003 | 0.339 |
| Miscellaneous | 0.000 | 0.003 | 1.000 |
| Professional | −0.001 | 0.005 | 0.900 |
| Socio-cultural | 0.000 | 0.003 | 1.000 |

Only the positive or negative character of each significant keyword's association with influence is given in Table 22. It was decided to show only the signs of the coefficients rather than their values, because it is difficult to interpret coefficient values modified by applying the Lasso method. However, the method retains the signs of the coefficients, which makes it relevant to show their signs.

The PageRank model (see Table 22) summarizes the most significant keywords and the role they play in deciding which theme group a particular author is most likely to belong to. Both Attention and Nominalization were correlated with authors belonging to the low influence group, suggesting that those who write about these two topics tend to end up with low influence as measured by PageRank. This finding coincides with the earlier theme profile analysis, where both Cognitive (the theme for Attention) and Language (the theme for Nominalization) issues were positively correlated with the low influence group. At the other end of the spectrum, Theory was positively correlated with authors in the high influence group, suggesting that scholars writing in that vein were very likely to receive high PageRank scores, a finding in line with the earlier one that Miscellaneous themes are positively correlated with the high influence group—Theory belongs to the Miscellaneous category.

**Table 22  PageRank keyword profile regression.**

| Keyword | Low/mid/high group | Positive/negative association | Theme group |
|---|---|---|---|
| Attention | Low | Positive | Cognitive |
| Nominalization | Low | Positive | Language |
| Extralinguistic information | Medium | Negative | Socio-cultural |
| Interpreters roles | Medium | Negative | Socio-cultural |
| Interpreting process | Medium | Negative | Cognitive |
| Schemata | Medium | Negative | Cognitive |
| Coping tactics | Medium | Positive | Training |
| Theory | High | Positive | Miscellaneous |
| Case studies | Low | Negative | Training |
| Domestication | Low | Negative | Language |
| Faithfulness | Low | Negative | Language |
| Fidelity | Low | Negative | Training |
| Cultural awareness | Medium | Negative | Language |
| Interpreting process | Medium | Negative | Training |
| Content validity | Medium | Positive | Language |
| Faithfulness | Medium | Positive | Professional |
| Interpreting styles | Medium | Positive | Language |
| Logic | Medium | Positive | Training |
| Non-standard expressions | Medium | Positive | Miscellaneous |
| Rights | Medium | Positive | Socio-cultural |
| Tem8 | Medium | Positive | Language |
| Thematic progression | Medium | Positive | Language |
| Certification | Medium | Positive | Socio-cultural |
| Non-linguistic context | Medium | Positive | Professional |
| Rhetoric fuzziness | Medium | Positive | Language |
| Textual coherence | Medium | Positive | Language |
| Copyright | Medium | Positive | Miscellaneous |
| Self evaluation | Medium | Positive | Training |
| Chinese classics | Medium | Positive | Language |

The second model (see Table 23) indicates that 20 keywords were significantly associated with the In-Degree measure. The largest effect was that Language-related words were more likely to be found in the Middle group, which coincides with the theme profile analysis. However, the Lasso regression was not able to detect that Socio-cultural themes predicted placement in the High In-Degree group. A possible explanation for this is that many Socio-cultural words were not significant on their own but collectively contributed to boosting an author into the high influence group.

The third model (see Table 24) indicates that three keywords were significantly associated with the Out-Degree measure. As found in both the theme profile cutoff and linear regressions, Cognitive words were positively correlated with being in the High Out-Degree group. Unlike the theme profile regressions, the model found that a Professional word correlated with the High group but it did not pick up on the impact of Language words.

**Table 23 In-degree keyword profile regression.**

| Keyword | Low/mid/high group | Positive/negative association | Theme group |
|---|---|---|---|
| Case studies | Low | Negative | Training |
| Domestication | Low | Negative | Language |
| Faithfulness | Low | Negative | Language |
| Fidelity | Low | Negative | Training |
| Cultural awareness | Medium | Negative | Cognitive |
| Interpreting process | Medium | Negative | Training |
| Content validity | Medium | Positive | Language |
| Faithfulness | Medium | Positive | Professional |
| Interpreting styles | Medium | Positive | Language |
| Logic | Medium | Positive | Training |
| Non-standard expressions | Medium | Positive | Miscellaneous |
| Rights | Medium | Positive | Socio-cultural |
| Tem8 | Medium | Positive | Professional |
| Thematic progression | Medium | Positive | Language |
| Certification | Medium | Positive | Socio-cultural |
| Non-linguistic context | Medium | Positive | Professional |
| Rhetoric fuzziness | Medium | Positive | Language |
| Textual coherence | Medium | Positive | Language |
| Self-evaluation | Medium | Positive | Training |
| Chinese classics | Medium | Positive | Language |

**Table 24 Out-degree keyword profile regression.**

| Keyword | Low/mid/high group | Positive/negative association | Theme group |
|---|---|---|---|
| Attention | Low | Negative | Cognitive |
| Sensory memory | High | Positive | Cognitive |
| Trade association | High | Positive | Professional |

## CONCLUSION

The present study demonstrates that CIS authors do indeed form discrete clusters among themselves, but the norms that usually govern the clustering of Liberal Arts and Natural/Empirical Sciences scholars (*Moser-Mercer, 1994*; *Gile, 2005*) cannot be used for classifying these groupings. Close examination of the citation data revealed that the majority of members of each cluster displayed one or more of the following defining characteristics: (1) their areas of research were similar; (2) they were influenced by the same theory; (3) they authored or cited 'classic' textbooks that contain the established fundamentals of the subject. All these characteristics indicate that the CIS community is a diverse one, with scholars forming into groups based on their shared characteristics. Despite this diversity, a small number of individuals stood out as the most influential. While the top 30 Western authors exerting the most influence in CIS had a wide range

of distinct areas of interest and expertise, all but one of the Chinese top 30 specialized in research into interpreting. It is also worth noting that a substantial proportion of the overall total of researchers had several of their works cited, while a minority was influential thanks to only one or two publications.

This paper also contributes to better understanding how research topics are associated with a CIS author's influence. It concluded that authors writing about non-mainstream topics (i.e., Miscellaneous themes) were more likely to be found in the high-influence PageRank group than those tackling 'bread-and-butter' subjects, and those writing on Socio-cultural themes were more likely to be placed in the high-influence In-Degree group. The study also identified several keywords significantly correlated with an author's network measures: Theory (high-influence PageRank group); Attention and Nominalization (low-influence PageRank); Language-related keywords (middle-ranking In-Degree); and Sensory Memory and Trade Association (high-ranking Out-Degree). The findings for keywords were broadly in line with those for themes, suggesting that authors who wish to make their mark in the academic community would do well to embrace certain topics while avoiding others.

When Interpreting Studies was in its infancy in the 1960s only a handful of isolated authors, scattered throughout the world, were conducting research (*Gile, 2013a*); today, despite its still relative youth in comparison with 'old timers' such as linguistics, philology, etc., it is well on the way to becoming a mature discipline in its own right (*Moser-Mercer, 2011*), and China's contribution to its rapid development has undoubtedly been considerable. The aim of this scientometric survey has been to provide a panorama of the evolution of Chinese Interpreting Studies while demonstrating the merits of blending traditional citation analysis with Social Network Analysis to produce such a survey. It is hoped that its findings might help authors better appreciate the trade-offs they need to make when choosing research topics and the potential academic influence that may result from their choices, as well as, more importantly, offering policy-makers new insights and food for thought as they chart the future course of CIS research.

## ACKNOWLEDGEMENTS

The authors wish to thank Ewan Parkinson for patiently reviewing multiple versions of this paper and for generously providing detailed suggestions on improving its quality; the authors would also like to express their gratitude to Faraz Zaidi for his valuable input on social network analysis.

### Funding

There was no external funding for this project. All the financial resources needed to carry out this research were contributed by one of the authors, Ziyun Xu.

### Competing Interests

The authors declare there are no competing interests.

## Author Contributions

- Ziyun Xu conceived and designed the experiments, performed the experiments, analyzed the data, contributed reagents/materials/analysis tools, wrote the paper, prepared figures and/or tables, performed the computation work, reviewed drafts of the paper, collected data and provided funding for this research project.
- Leonid B. Pekelis conceived and designed the experiments, performed the experiments, analyzed the data, contributed reagents/materials/analysis tools, performed the computation work, reviewed drafts of the paper.

## Data Availability

The following information was supplied regarding the deposition of related data:
GitHub: https://github.com/danielxu85/CIS.

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
