# Peer review of "A survey of Chinese interpreting studies: who influences who … and why?"

_PeerJ Computer Science, doi:10.7717/peerj-cs.14_

## Round 0.1 · original submission · Major Revisions

· Academic Editor

Major Revisions

In general, the evaluation received by the paper has been quite positive. Nevertheless, one of the reviewers has concerns about the description of the tools used in the analysis. Specifically, authors should provide information about the software and methods used to obtain the different networks and metrics. The rest of the comments are related to minor changes, although I encourage authors to incorporate them in the new version go the paper.

Reviewer 1 ·

Basic reporting

The aim of this paper it to perform a deep bibliometric analysis of the Chinese Interpreting Studies using social network analysis techniques.

In general, the paper is well written and organized. Although, the main drawback is that the paper is biased to the bibliometric Chinese literature. I miss a lot of important reference to co-word, co-citation, co-author, etc. In fact, the map of a research field is also known as Science Mapping analysis.

Moreover, the paper is somewhat large and some parts of it are repeated constantly along the paper.

Experimental design

The data acquisition is not well explained. Authors should make an effort, an explian the procedure and query used.

Validity of the findings

The findings are correct.

Additional comments

In what follows, some suggestions are comments are listed:

Line 77. Includes reference to social network analysis and citation analysis

Line 201. How data acquisition has been performed? Authors should describe the procedure and the query used to retrieve the data.

Line 295. “Cited one another’s research” is also known as co-author citation.

Line 317. Why the authors not use h-index to measure the most influential scholars?

Line 343. The authors used a non-common similarity measure. According to van Eck 2009, the Association strength should be used to normalize co-occurrence network. Also, the equivalence index has been traditionally used in co-word network.

Line 367. Under my point of view, 40% of authors not-matched is too much.

Line 394. Authors may refer to Zipf’s law: “Zipf's law states that while only a few words are used very often, many or most are used rarely”.

Line 462. The software used to perform the analysis and visualization should me mentioned. There are a great variety of software tool to perform a science mapping analysis (e.g. VOSViewer, SciMAT, Sci2 Tool, VantagePoint, etc.), and also a great variety of software to visualize network (e.g. Gephi, Pajek, etc.)

Line 512. Figure 3 should be enlarged and better laid out. Authors should draw the network avoiding use circular layout. Now, it is very difficult to see the relation between nodes of the same cluster.

Line 606. H-Classics should be used to highlight the mots influent works.

Reviewer 2 ·

Basic reporting

The paper carries out a scientometric survey on Chinese Interpreting studies (CIS) to obtain a description of the ways in which CIS scholars interact and which are the most influential studies. They use both traditional technique of citation analysis and Social Network Analysis.

The paper describes correctly their purpose and a considerable number of works are included. Moreover, different metrics are considered in the study. However, some points should be improved related to disscussion of the results obtained.

Experimental design

It would be necessary that authors detail the software and methods used to obtain results. Concretely,

+ In section 6.1, it is not mentioned a software to carry out the work. How authors have obtained the different networks and their different metrics?

+ In section 6.3, the specification of the software and the methods used to carry out this study neither they are mentioned.

Also, it would be convenient that authors specify the number of years considered in the study.

Validity of the findings

It is appropriate

Additional comments

Authors should improve the next points:

- Figure 3 it is very complicated to visualize. It would be convenient that authors try to simply or divide it to be able to understand better the results.

- In general, the section 6.3 would be reviewed. Some times it is very difficult to understand appropriately the results obtained. Definitions, descriptions and discussions are mixed and sometimes information is duplicated. It would be interesting that authors try following the same steps in the discussion of all results. First, it would be necessary to comment the specific numerical results of each table (in some tables, it is not specified the metrics shown neither in this section neither in section 5). Second, it would be necessary to show specific examples and detail the general conclusions. Normally, all information is available, but they should be restructured.

- Finally, some times authors use an acronym, but this is specified later. For example:
- - Line 51 the acronym “BCE” it is not specified previously.
- - Line 260 the acronym “PRA” it is not specified previously.

---

## Round 0.2 · accepted · Accept

· Academic Editor

Accept

After this second round of reviews, the paper has considerably improved, so it has been accepted for publication.

Reviewer 1 ·

Basic reporting

Authors have addressed the majority of the changes suggested. Therefore, I recommend to accept the paper.

Experimental design

Ok

Validity of the findings

Ok

---

## Author Rebuttal · Round 0.2

Rebuttal letter

Dear Dr. Ventura:

Thank you very much for providing me with the valuable comments from the readers. They have proved most helpful in improving the overall quality of this study. I have substantially revised the paper to address their concerns. In particular, the software application used to analyze and visualize citation data was revised, and more detail on the analytical approach used was provided. Reviwers asked for more information about the source of the data; I deposited in GitHub the citation data used for this study and the codes written for statistical analysis and visualization, and they are accessible from this link: https://github.com/danielxu85/CIS

I have addressed the comments by the reviewers point by point below. I believe that the revised paper now meets the criteria for publication by PeerJ Computer Science. Please do not hesitate to contact me if you have any additional questions or concerns. Thank you again for your time and consideration.

Sincerely,
Ziyun Xu
On behalf of all authors

Reviewer 1
*In general, the paper is well written and organized. Although, the main drawback is that the paper is biased to the bibliometric Chinese literature.*
Response: The purpose of this study to paint a panorama of Chinese Interpreting Studies (CIS) using techniques from Computer Science and assess the academic influence of the major players in the field. Every effort was made towards amassing a comprehensive collection of data. While a significant number of papers were written in Chinese, they nonetheless contain English bibliographic references, and over 60% of the MA theses in the data-set were written in English.

This provides a good balance of English and Chinese citation data, and serves to illuminate how the influences exerted by Chinese and Western CIS scholars differ.

*I miss a lot of important reference to coword, cocitation, coauthor, etc. In fact, the map of a research field is also known as Science Mapping analysis.*

Response: We believe that these comments are related to the data visualization and keyword correlation portions of the study. Specific revisions and clarifications are made later in this letter (see the responses under lines 343, 462 and 512 for more detail). Briefly, heeding the reviewer's suggestion, we redeveloped the data visualization methodology and explained why author-generated groupings of keywords were favored over association strength and equivalence index.

*Moreover, the paper is somewhat large and some parts of it are repeated constantly along the paper.*

Response: The paper has been edited to eliminate overlap between these different sections, and to ensure that readers can follow the thread and appreciate why and how the study was conducted. Originally, for the sake of clarity, the research questions were repeated in the Methodology and Results section. To reduce redundancy, a much-shortened version of each question was presented in the different sections, rather than repeating the exact same question over and over. For example, the shorthand phrase 'author interaction' was used to refer to Question 1 in the Methodology and Results and Discussions sections, so the repetition is eliminated, and the readers still see consistent signposts to help them understand how different sections work together to address the main questions in the study.

*The data acquisition is not well explained. Authors should make an effort, and explain the procedure and query used.*

Response: We agree with this comment. To address it, the data organization section was retitled 'data collection and organization'. In the revised section 4.2, we discuss the acquisition of data from field trips to university libraries, interlibrary loans, book purchases, and academic databases

such as CNKI, Wanfang and the National Digital Library of Theses and Dissertations in Taiwan. We then describe manual data entry ; and the creation of a schematic graph to illustrate how the data was organized (see Figure 1).

*Line 77. Includes reference to social network analysis and citation analysis*
Response: I have added references to this section. I have added references (Baumgartner & Pieters, 2003) and (Otte & Rousseau, 2002) to this section.

*Line 201. How data acquisition has been performed? Authors should describe the procedure and the query used to retrieve the data.*
Response: As described in my response to the previous comment, a detailed description of the data collection and assimilation procedure was added to section 4.2. It was clarified that the data was organized using the idea behind SQL, but the data was actually manually entered into Excel Spreadsheets, so no query was used. The raw data was deposited into GitHub, and can be viewed from this link: https://github.com/danielxu85/CIS

*Line 295. "Cited one another's research" is also known as coauthor citation.*
Response: Agreed, and a co-author citation reference (Newman, 2001) was added to page 14 of the revised manuscript.

*Line 317. Why the authors not use h-index to measure the most influential scholars?*
Response: Similar to Impact Factor (IF), H-Index is a first-order centrality measure in assessing an author's academic influence (Wasserman & Faust, 1994). It has some of the same limitations as the IF score in that it does not account in any way for the secondary influence of the source paper. A citation in a paper that never receives any citations should not be considered the same as a citation in a groundbreaking, highly-cited work. Because of this, PageRank, a higher-order centrality measure, was favored over H-Index as the primary method for assessing the research impact of scholars in this study: unlike IF or H-Index, PageRank allows me to account for secondary influences of CIS authors in this study. While it was initially developed by Google's

founders to rank web pages, a number of previous studies (see for example Ding, Yan, Frazho, & Caverlee, 2009; Ma, Guan, & Zhao, 2008; Waltman & Yan, 2014) have demonstrated that it can be reliably applied in the analysis of citation data.

*Line 343. The authors used a non-common similarity measure. According to van Eck 2009, the Association strength should be used to normalize cooccurrence network. Also, the equivalence index has been traditionally used in coword network.*

Response: While the reviewer does bring up an interesting suggestion for analyzing the similarity of authors in our citation network, to do so would not exactly coincide with the research question, which concerned how the research topics authors choose correlate with their influence.

To answer this question a measure of author influence was needed for the dependent or response variable. Since authors cite each other when they believe the source of the citation has useful information or is well-regarded, a citation network is a useful means of measuring influence. In this sense a citation network is not unlike the Internet in which websites receive numerous links if they are seen as authoritative hubs of knowledge. Common measures of website influence are those which we computed: degree centrality, PageRank and eigenvector centrality.

If we had used the method suggested by the reviewer, and followed it with a clustering analysis, this would have resulted in our measuring the correlation of research topics to co-citation clusters. In other words, this would have answered the research question "Do authors who tend to cite each other also tend to research similar topics?"

Another option would have been to use association strength to cluster the independent variables, i.e. keyword counts. By adopting this approach we could have clustered authors according to groups of similar keywords they used. This would have provided a data-driven grouping of research topics which we might then have used in a regression on our measure of influence. We

agree with the reviewer that this would have been an interesting direction to explore, but it was decided instead to generate our own groupings of keywords into research topics and use the keyword profiles as the independent variables for this paper.

This was decided on for two reasons. Firstly, author-generated groupings of keywords are guaranteed to be semantically meaningful, while a clustering of keyword profiles is not. This seemed like the best approach for aligning broad categories of research topics with author influence and answering such questions as "Do authors who write more cognitive papers tend to cite more?"

Secondly, we wanted to find the correlation of individual keywords to author influence— by clustering them instead we might have failed to detect some correlations. For example, a keyword which was correlated with author influence but not particularly associated with any keyword cluster would not have surfaced if we had first clustered keywords by association strength then regressed them on influence.

The above arguments apply equally well to the reviewer's suggestion to use equivalence index, given that that too measures the similarities between keywords, used to cluster them into groups.

We agree with the reviewer that a data-driven approach to finding keyword clusters, perhaps by using a combined clustering-regression procedure, would be an interesting and promising direction for future work, but feel that the results of our analysis of the correlation between author influence and author-generated individual and grouped keywords are of themselves a useful contribution to understanding the extent to which certain topics are valued more highly than others in the field of CIS.

*Line 367. Under my point of view, 40% of authors not-matched is too much.*
Response: This was just a property of the data, and has nothing to do with the validity of the study. Details were added in section 5.3 to further clarify this point in the revised paper. Because

of China's unique intellectual traditions in the early stages of CIS' development as an academic discipline the overwhelming majority of papers published had no bibliographic references. In addition, most of them were self-reflective writings based on anecdotal evidence, rather than data-driven research, so citations were not really necessary. However, these early studies were included in the data-set for three reasons: firstly, these articles were produced during CIS's initial stage, as per Schneider's model of the development of scientific disciplines (2009); secondly, many of them received citations from later studies, which indicated that they served as the foundation for the development of CIS and brought academic value to the field; and finally, their lack of citation data does not affect the validity of the findings for this research question – despite having no outgoing citations, the authors of these early studies received incoming citations from later scholars, their academic influence being documented in the data-set.

*Line 394. Authors may refer to Zipf's law: "Zipf's law states that while only a few words are used very often, many or most are used rarely".*

Response: The reviewer's suggestion is highly appreciated and  Zipf law is certainly applicable to how the frequencies of author's connections are distributed.   While the Gini coefficient calculation in the present study demonstrated that CIS authors' influence measures were unevenly distributed, this has nothing to do with the unequal distribution of author citations. The sentences in question refer to the allocation of the authors into different influence groups and not their connections following Zipf's Law. It was emphasized that depending on the network measure selected, the groups were divided by different cutoff points, and that these cutoffs were determined in a data-dependent manner.

*Line 462. The software used to perform the analysis and visualization should me mentioned. There are a great variety of software tool to perform a science mapping analysis (e.g. VOSViewer, SciMAT, Sci2 Tool, VantagePoint, etc.), and also a great variety of software to visualize network (e.g. Gephi, Pajek, etc.)*

Response: The reviewer makes a very good point, which is shared by the second reviewer. These comments have been addressed in the revised paper. A new software application, Tulip, which

was designed to handle large complex networks (see for example Suderman & Hallett, 2007), was adopted to replace the original analysis conducted using Gephi. As pointed out by the first reviewer, some software applications are mainly for scientific mapping analysis while others are used for visualization. The advantage of Tulip is that it is capable of both analyzing and visualizing relational data, because it has a number of layout and clustering algorithms in addition to network metrics. A step-by-step explanation has been provided of how Tulip was used to calculate the PageRank scores of CIS authors and generate the graphs in section 5.1.

*Line 512. Figure 3 should be enlarged and better laid out. Authors should draw the network avoiding use circular layout. Now, it is very difficult to see the relation between nodes of the same cluster.*

Response: We agree with this suggestion, and have re-developed figure 3 with it in mind. The new graphs were created using a force-directed algorithm called Fast Multipole Multilevel Method (Hachul & Jünger, 2005). This algorithm places close together nodes that are multiply connected to each other while distancing those that are not directly connected. This improved design makes it easy to identify the community structures of large networks such as that of CIS, which contains over 12,000 nodes and 50,000 edges. In addition, the edges were rendered invisible in the graphs to ensure that nodes are clearly visible to the readers.

*Line 606. H-Classics should be used to highlight the most influential works.*

Response: The reason for not using H-Index was elaborated on in my response to Line 317; briefly, it does not consider secondary influence of authors. However, this comment did bring to my attention that inconsistent metrics were used throughout the paper. In the interest of standardizing the use of metrics throughout the whole study, PageRank values were used rather than using In-Degree scores to rank papers in the revised version.

Reviewer 2:

*It would be necessary that authors detail the software and methods used to obtain results. Concretely, + In section 6.1, it is not mentioned a software to carry out the work. How authors have obtained the different networks and their different metrics?*

Response: We agree with this comment. We have detailed our revisions on this issue in the response to Line 462 from Reviewer 1; briefly, we have used a new software application, Tulip, to redevelop the graph, and detailed the procedure for the visualization.

*Figure 3 it is very complicated to visualize. It would be convenient that authors try to simply or divide it to be able to understand better the results.*

Response: We agree with this comment. This issue was covered by the first reviewer: my response can be found under Line 512 above; in short, a force-directed algorithm was adopted in the revision to bring together nodes that are multiply connected, so it would be much easier for the readers to understand the graph.

*In general, the section 6.3 would be reviewed. Sometimes it is very difficult to understand appropriately the results obtained. Definitions, descriptions and discussions are mixed and sometimes information is duplicated. It would be interesting that authors try following the same steps in the discussion of all results. First, it would be necessary to comment the specific numerical results of each table (in some tables, it is not specified the metrics shown neither in this section neither in section 5). Second, it would be necessary to show specific examples and detail the general conclusions. Normally, all information is available, but they should be restructured.*

Response: To address this, our solution was to add a subsection titled Summary of Significant Findings to the beginning of section 6.3. This summary section (6.3.1) is designed to contextualize the results and their significance. More specifically, the summary begins with a brief recap of the low, mid, and high grouping divisions. A table was added listing all the significant coefficients from the multinomial regressions in an easy-to-read format, and explanations were provided on how to read the tables. Lastly, the most significant findings from this analysis were summarized to inform readers which topics are written about by the most

influential authors. The summary is followed by a detailed discussion of the results in subsections 6.3.2-6.3.5.

*Line 51 the acronym "BCE" it is not specified previously.*

Response: BCE stands for Before Common Era, and can be used interchangeably with BC ('Before Christ'). However, BCE is often preferred to BC because it avoids the explicit reference to Christianity. The abbreviation is used commonly enough in English writing for there to be no need to spell it out.

*Line 260 the acronym "PRA" it is not specified previously.*

Response: Agreed. PRA is a shorthand for PageRank Algorithm. The abbreviation is properly introduced in the revised text on page 14.